# Skillful multiyear predictions of ocean acidification in the California Current System

Riley X. Brady [1✉], Nicole S. Lovenduski [1], Stephen G. Yeager [2], Matthew C. Long [2] & Keith Lindsay[2]

The California Current System (CCS) sustains economically valuable fisheries and is particularly vulnerable to ocean acidification, due to its natural upwelling of carbon-enriched waters that generate corrosive conditions for local ecosystems. Here we use a novel suite of retrospective, initialized ensemble forecasts with an Earth system model (ESM) to predict the evolution of surface pH anomalies in the CCS. We show that the forecast system skillfully predicts observed surface pH variations a year in advance over a naive forecasting method, with the potential for skillful prediction up to five years in advance. Skillful predictions of surface pH are mainly derived from the initialization of dissolved inorganic carbon anomalies that are subsequently transported into the CCS. Our results demonstrate the potential for ESMs to provide skillful predictions of ocean acidification on large scales in the CCS. Initialized ESMs could also provide boundary conditions to improve high-resolution regional forecasting systems.

[1] Department of Atmospheric and Oceanic Sciences and Institute of Arctic and Alpine Research, University of Colorado, Boulder, CO 80309, USA. [2] Climate and Global Dynamics Laboratory, National Center for Atmospheric Research, Boulder, CO 80305, USA. ✉email: riley.brady@colorado.edu

Ocean acidification is an ongoing large-scale environmental problem, whereby the absorption of anthropogenic $CO_2$ by the ocean lowers its pH, impacting ocean ecosystems worldwide[1]. The California Current System (CCS) supports productive fisheries crucial to the US economy and is particularly vulnerable to ocean acidification due to the upwelling of naturally corrosive (i.e., relatively low pH) waters to the surface[2]. The upwelling process results from equatorward winds along the western North American coastline. These winds facilitate both coastal upwelling and curl-driven Ekman suction, forcing waters enriched in carbon and nutrients from beneath the thermocline to the surface[3]. These nutrient subsidies drive high productivity in CCS waters, essential to supporting regional fisheries[4]. However, the upwelled waters are also corrosive due to their high remineralized carbon content. The air-to-sea flux of anthropogenic $CO_2$ into the CCS further compounds this natural acidity. Multiple studies over the past decade have observed coastal CCS waters that are anomalously low in surface pH relative to the historical state of the system and undersaturated with respect to calcium carbonate minerals[5–7]. These conditions adversely affect a wide range of organisms that precipitate calcium carbonate shells, such as pteropods, coccolithophores, and shellfish[1]. Shellfish in particular contribute significantly to the $6B in revenue per year provided by commercial and recreational fisheries in the CCS[8]. The CCS's intersection between economically valuable fisheries and natural vulnerability to ocean acidification makes it a high-priority region to study for multiyear biogeochemical predictions.

Prediction efforts for the CCS have focused primarily on using seasonal forecasts of sea surface temperature[9–11] (SST) and biogeochemical variables[12] (e.g., dissolved oxygen and bottom pH) as inputs into ecosystem forecasting models. A more recent effort demonstrates the potential for skillful initialized predictions of surface chlorophyll in the CCS with 2 year forecasts[13]. However, no studies have attempted to predict ocean biogeochemistry in the CCS at the multiannual to decadal scale, as decadal forecasting of ocean biogeochemistry is still in its infancy[14–18]. This temporal scale is critical for fisheries managers, as it aids them in setting annual catch limits, changing and introducing closed areas, and adjusting quotas for internationally shared fish stocks[19]. Some level of skill can be provided by persisting anomalies from year-to year in the system[19]. These so-called persistence forecasts are commonly used as a reference to put initialized skill into context and work at lead times commensurate with the decorrelation timescales of the system[9–11,19]. On the other hand, initialized predictions use a physically based modeling framework to advance information from initial conditions forward in time; if the system is predictable (i.e., sufficiently deterministic) and the model skillful, this can yield a powerful forecasting framework. Ensemble simulations of initialized ESMs provide the most powerful approach currently available for improving upon decadal persistence forecasts. Their coupling of global physical models of the atmosphere, ocean, cryosphere, and land with the carbon cycle, terrestrial and marine ecosystems, atmospheric chemistry, and natural and human disturbances allows one to deeply investigate how interactions between the physical climate system and biosphere lead to predictability in a complex system such as the CCS[20]. These predictions have the potential to improve upon persistence forecasts, pushing the horizon of forecasting ecosystem stressors past a single season or year.

Here we use an initialized global ESM with embedded ocean biogeochemistry, the Community Earth System Model Decadal Prediction Large Ensemble[21] (CESM-DPLE), to assess retrospective forecasts of surface pH anomalies in the CCS from 1955 through 2017. We find that CESM-DPLE has the potential to predict surface pH anomalies for up to 5 years in advance in some regions of the CCS, and already exhibits skill out to 1 year in advance relative to historical observations. Predictability in surface pH results mainly from the initialization of dissolved inorganic carbon (DIC) anomalies, which are subsequently advected into the CCS, modifying local pH conditions.

## Results

**Experimental approach.** The CESM-DPLE employs an ocean model with nominal $1° × 1°$ horizontal resolution and 60 vertical levels. Forty ensemble members were initialized annually on November 1st from a forced ocean-sea ice reconstruction (hereafter referred to as the reconstruction) and then the coupled simulations were integrated forward for 10 years (Fig. 1a, b; see Model Simulations and Drift Adjustment in Methods section). The reconstruction is skillful in representing surface pH variability on seasonal to interannual timescales in the CCS (Fig. 2). Due to the diverse terminology used in weather and climate forecasting[22], we are careful with our definitions. We use the phrase potential predictability when referring to correlations between CESM-DPLE and the reconstruction. High correlation coefficients (i.e., high potential predictability) represent the theoretical upper limit for predictions in the real world, given the chaotic nature of the climate system[23]. We use the phrase predictive skill when comparing CESM-DPLE to observations; skill demonstrates our ability to predict the true evolution of the real world with CESM-DPLE. We quantify our ability to predict anomalies with the anomaly correlation coefficient (ACC), and our accuracy in predicting anomaly magnitudes with the normalized mean absolute error (NMAE; see Statistical Analysis in Methods section). We compare our initialized forecasts to a

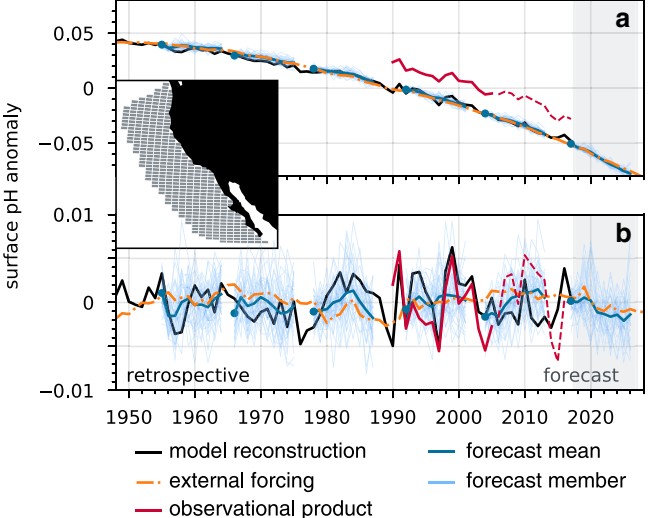

**Fig. 1 Experimental design of the decadal prediction system. a** Trended and **b** detrended, area-weighted annual surface pH anomalies for the (black) reconstruction, (red) observational product, (orange) Community Earth System Model Large Ensemble (CESM-LE) ensemble mean, and (blue) CESM Decadal Prediction Large Ensemble (CESM-DPLE) decadal forecasts initialized in 1954, 1965, 1977, 1991, 2003, and 2017 (other initializations were omitted for visual clarity). The dark blue line is the ensemble mean forecast, and thin blue lines are the individual 40 forecasts. The blue dots do not sit exactly atop the black line due to the rapid divergence of forecasts away from initialization within weeks. The dashed red lines denote when the model loses observed variability in atmospheric $CO_2$ forcing (Supplementary Fig. 1A). The inset shows the California Current Large Marine Ecosystem bounds, over which all area-weighted analyses are computed.

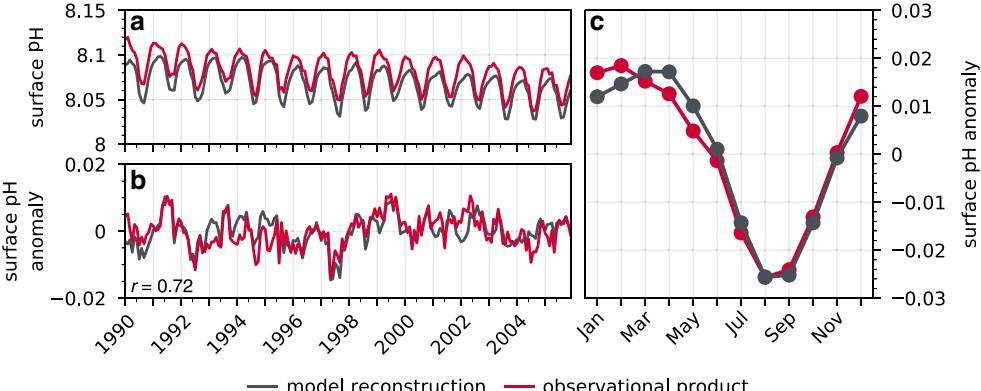

**Fig. 2 Area-weighted temporal evaluation of surface pH. a** Monthly surface pH in the California Current over 1990–2005 for the model reconstruction (black) and observational product (red). **b** As in **a**, but for anomalies after removing a second-order polynomial fit and the seasonal cycle. The correlation coefficient between the observational product and model reconstruction is shown in the bottom left of **b**. **c** As in the other panels, but for the mean monthly seasonal cycle over 1990–2005.

simple persistence forecast and the uninitialized CESM Large Ensemble[24] (CESM-LE) mean, which includes the same external forcing (i.e., rising atmospheric $CO_2$) as the CESM-DPLE. The former assesses whether CESM-DPLE is useful relative to a simple forecasting method, while the latter determines the degree to which initialization engenders predictability beyond that afforded by supplying the model with time-varying forcing. We test predictive skill by comparing the initialized forecasts to a gridded observational product of surface pH from the Japan Meteorological Agency (JMA), which spans 1990–2017[25,26]. This product is based upon empirical relationships derived for alkalinity and $pCO_2$ as functions of in situ measurements, such as SST and sea surface height, which were then used in a carbonate system solver to derive gridded surface pH (see Observational Product in Methods section). Our focus in this study is on surface pH anomalies within the California Current Large Marine Ecosystem (see the inset in Fig. 1 for the spatial domain). We focus on the entire Large Marine Ecosystem, since the 1° × 1° model grid cannot resolve the coastal upwelling of corrosive waters that occurs on scales smaller than the grid resolution. We remove a second-order polynomial fit from all surface pH time series, since the long-term ocean acidification signal dominates over the 1955–2017 hindcast period (Fig. 1a). We aim to test our ability to predict year-to-year variations in CCS surface pH anomalies (Fig. 1b), which act to temporarily accelerate or slow down the ongoing ocean acidification trend.

**Model evaluation.** Previous evaluations of the physical circulation and carbonate chemistry in the version of CESM used for CESM-DPLE suggest that, despite the relatively coarse 1° × 1° model grid, CESM provides a good fit to observational climatologies of alongshore wind stress, surface $pCO_2$, and air–sea $CO_2$ fluxes in the CCS[27,28]. Modeled alongshore wind stress—the primary driver of coastal upwelling—closely matches the magnitude and seasonality of observations, with peak upwelling-favorable conditions spanning April to September[27]. The large-scale spatial structure of air–sea $CO_2$ fluxes in the model exhibits poleward $CO_2$ uptake and equatorward $CO_2$ outgassing, matching that of modern observationally based estimates[28,29]. Importantly, we note that CESM cannot capture the nearshore outgassing of $CO_2$ associated with the coastal upwelling of carbon-enriched waters that occurs on a scale smaller than the resolution of the model grid[28,30]. The modeled monthly climatology of area-weighted surface ocean $pCO_2$ in the CCS closely resembles that of the observationally based estimate, due to the

model's proper simulation of the magnitude and phasing of thermal (solubility-driven) and non-thermal (circulation- and biology-driven) $pCO_2$ effects[28,29].

We further evaluate the carbonate chemistry of the CCS region in CESM-DPLE by comparing surface ocean pH from our reconstruction with the gridded JMA observational pH product[25,26]. We limit the evaluation period to 1990–2005, as the JMA observational product begins in 1990, and the reconstruction is forced using non-historical atmospheric $CO_2$ from 2006 onwards (Supplementary Fig. 1). Over the 1990–2005 period, the spatial distribution of surface pH climatologies in the reconstruction closely match that of the observational product, with both suggesting higher surface pH during the wintertime downwelling season and lower surface pH in the summertime upwelling season (Supplementary Fig. 2, see also Fig. 2c). High-resolution model solutions demonstrate similar spatial patterns and seasonality of surface pH in this region[31]. The reconstruction has a slight acidic bias (Fig. 2a), with a relative mean bias in the hydrogen ion concentration ([H+]) ranging from 2.9% to 4.2% across the CCS (Supplementary Fig. 2, I to L). Over the area-weighted CCS (Fig. 2), the reconstruction simulates a linear change in surface pH of −0.026 over the 1990–2005 period, compared to the observational product's linear change of −0.029 (Fig. 2a). Both the reconstruction and observational product exhibit an interannual standard deviation of 0.003 in surface pH. Thus, the interannual variability in both the model and observations is between 1.5 and 2 times greater than the ocean acidification trend over the course of 1 year. Surface pH anomalies in both the reconstruction and observational product exhibit a decorrelation time scale of four months (Supplementary Fig. 3). The reconstruction closely replicates surface pH monthly anomalies (second-order polynomial fit and seasonal cycle removed) from the JMA observational product (Fig. 2b), with a linear correlation coefficient of 0.72.

We identify the drivers of reconstructed surface pH variability in the CCS by estimating the contributions from variations in salinity, alkalinity, SST, and DIC (see Linear Decomposition in Methods section). The two major terms driving variability in surface pH are DIC and SST, whose standard deviation is approximately three times that of surface pH (Supplementary Fig. 4). These two terms exhibit low-frequency variability and are significantly correlated with modes of variability such as the Pacific Decadal Oscillation (PDO) and El Niño–Southern Oscillation (ENSO). The linear correlation coefficient between DIC and SST residuals and the PDO is 0.66 and 0.73, and ENSO is 0.52 and 0.64, respectively (Supplementary Table 1). Since

 **3**

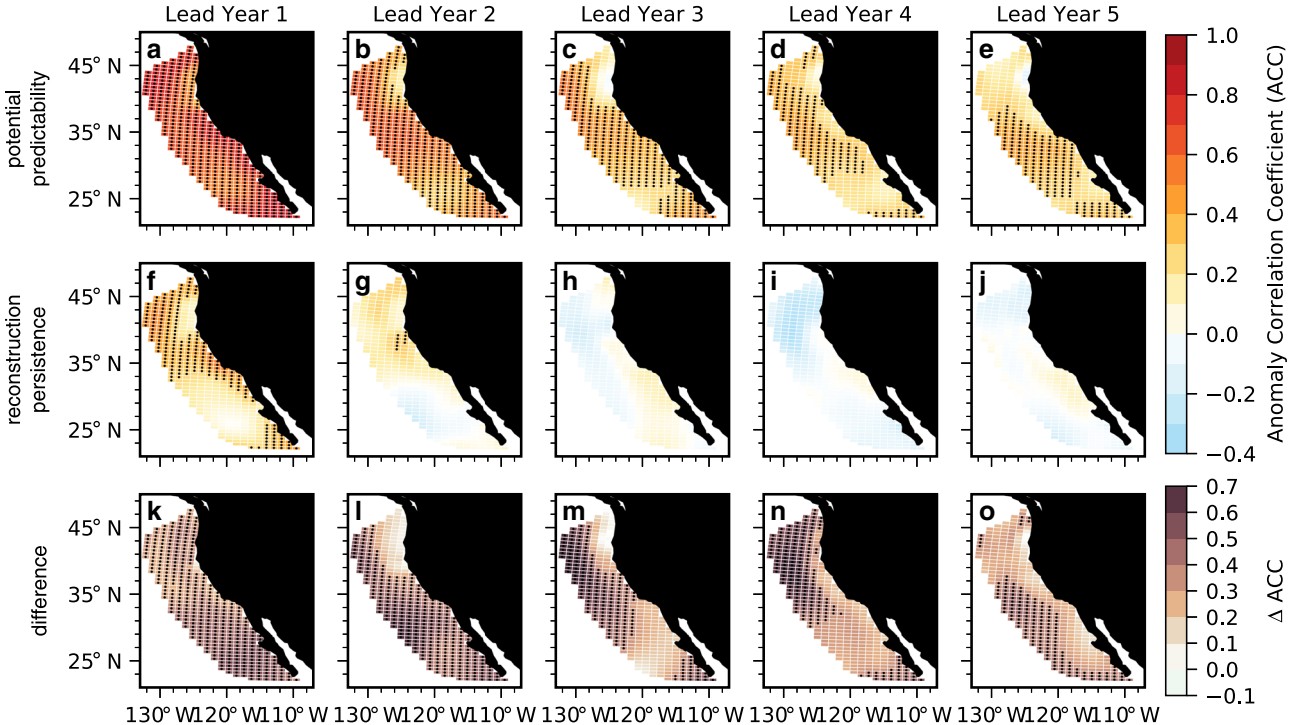

**Fig. 3 Potential predictability of surface pH anomalies. a–e** Anomaly correlation coefficients (ACCs) for Community Earth System Model Decadal Prediction Large Ensemble (CESM-DPLE) initialized forecasts of detrended annual surface pH anomalies for lead years one through five correlated with the reconstruction. **f–j** Persistence forecast for the reconstruction for lead years one through five. Stippling in **a–j** denotes statistically significant correlations at the 95% level using a *t* test. An effective sample size is used in the *t* test to account for autocorrelation in the two time series being correlated. **k–o** Difference between the CESM-DPLE forecast ACCs and persistence (ΔACCs). Stippling indicates that the initialized prediction is statistically significant over the persistence forecast at the 95% level using a *z* test. Only positive ACCs and ΔACCs are stippled.

surface pH is the small residual of many variables, it has a correlation coefficient of nearly zero with both modes of variability (Supplementary Table 1).

**Predictions of simulated and observed surface pH**. Retrospective forecasts of detrended annual surface pH anomalies in the CCS suggest a potential to predict surface pH up to 5 years in advance over a simple persistence forecast (Figs. 3 and 4). Although a persistence forecast is valuable at lead year 1 in parts of the CCS (Fig. 3f), the initialized forecast is statistically significant over persistence nearly everywhere (Fig. 3k). By lead year 2, persistence begins to yield negative ACCs in the southern portion of the CCS, while retaining some positive correlation in the north; ACCs become non-significant and weakly negative from lead year 3 and beyond (Fig. 3h–j). The initialized forecast, in contrast, retains predictability in the central and southern CCS through lead year 5 (Fig. 3a–e). Initialized predictions have higher ACCs (ΔACC) than a persistence forecast everywhere out to 5-year leads, save for three coastal grid cells along the coastal Pacific Northwest in lead year 3 (Fig. 3k–o). An area-weighted perspective of the CCS reveals that the initialized forecast is statistically significant over both persistence and the uninitialized forecast through 5-year leads (Fig. 4a). The lead year 1 ACC of 0.72 explains over 50% of the variance in predicted surface pH anomalies and is comparable or better than the skill achieved by seasonal forecasts of SSTs in the CCS[9,10]. The NMAE is smaller than both persistence and the uninitialized forecast and falls within the magnitude of surface pH interannual variability in the model reconstruction over all 10 lead years (Fig. 4b).

Because the reconstruction simulates the mean state, seasonal cycle, and variability of surface pH in the CCS well (Fig. 2 and Supplementary Fig. 2), potential predictability extends to

predictive skill relative to the observational product (Fig. 5). Initialized predictions have positive ACCs throughout most of the CCS at lead year 1 (Fig. 5a), and exhibit skill over persistence through lead year 4 from Cape Mendocino to Baja California (Fig. 5k–n). Persistence in the observationally based surface pH estimate is somewhat useful south of Cape Mendocino at lead year 1, but yields negative ACCs from lead years 2–5 throughout most of the CCS (Fig. 5f–j). Note, however, that none of these correlations are statistically significant at the 95% level. Across all 5 lead years, ACCs from the initialized predictions are larger than those of observational persistence for most of the CCS (Fig. 5k–o), with an area-weighted mean ΔACC (the difference between ACCs for the initialized ensemble and observational product) ranging from 0.04 to 0.43. Skill is lost for the southernmost portion of the CCS by lead year 2 (Fig. 5b), followed by the Pacific Northwest at lead year 3 (Fig. 5c). Mean absolute error in the initialized predictions of the observed surface pH is smaller than that of observational persistence for most of the CCS over 5 lead years (Fig. 6k–o), and primarily falls within the magnitude of surface pH interannual variability in the observations (Fig. 6a–e). Our results suggest that CESM-DPLE could be used for multiyear forecasting of surface pH variability in the CCS today.

**Mechanisms of surface pH predictability**. We are further interested in what lends predictability to surface pH in the CCS. We begin by investigating predictability in the driver variables of pH: temperature, salinity, DIC, and alkalinity. By scaling these variables to common pH units (see Linear Decomposition in Methods section), we can deduce which drivers aid the most in predicting surface pH. We find that predictability in salinity-normalized DIC (sDIC) has the largest influence on surface pH predictability over all 10 lead years (Fig. 4c). The combined predictability of both SSTs

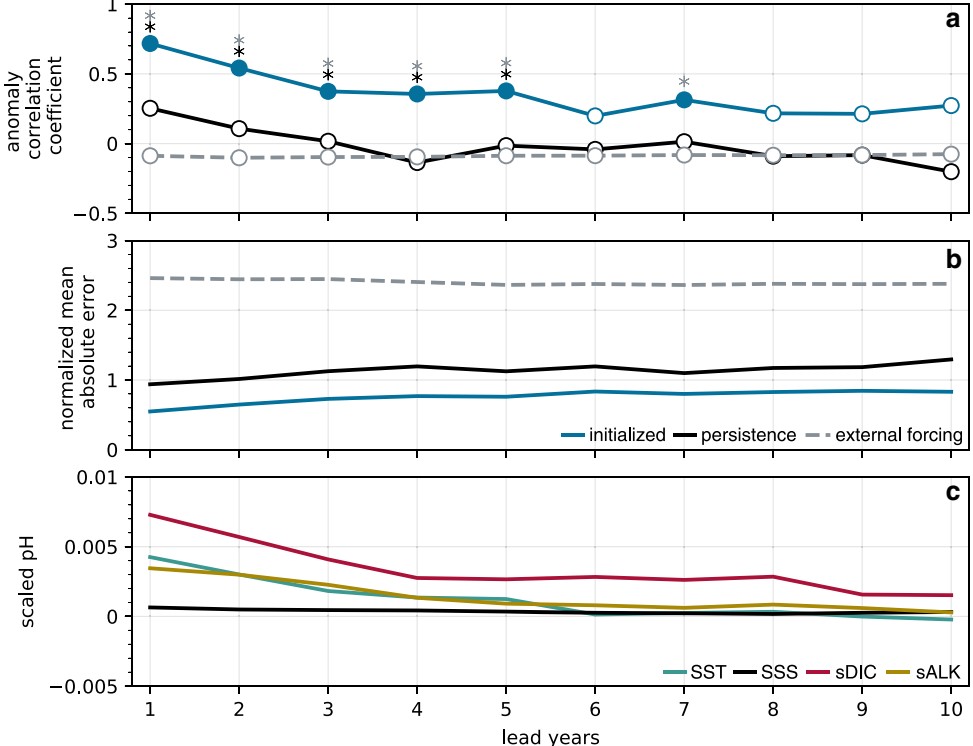

**Fig. 4 Domain-wide potential predictability of surface pH anomalies. a** Anomaly correlation coefficients (ACCs) for 10 lead years for (blue) the Community Earth System Decadal Prediction Large Ensemble (CESM-DPLE), (black) a persistence forecast from the reconstruction, and (gray) the uninitialized CESM Large Ensemble (CESM-LE) ensemble mean. Filled circles denote statistically significant positive correlations at the 95% level using a *t* test. An effective sample size is used in the *t* test to account for autocorrelation in the two time series being correlated. The critical value required for a statistically significant correlation ranges from 0.26 to 0.32 across leads, as computed by inverting the *t* statistic formula. Black and gray asterisks indicate significant predictability over persistence and the uninitialized forecast at the 95% level using a *z* test, respectively. **b** As in **a**, but for normalized mean absolute error (NMAE) and without significance testing. Values below (above) one indicate that the forecast falls within (outside of) the interannual variability of surface pH in the reconstruction. **c** Scaled predictability in common pH units (see Linear Decomposition in Methods) of (black) sea surface salinity (SSS), (teal) sea surface temperature (SST), (gold) salinity-normalized alkalinity (sALK), and (red) salinity-normalized dissolved inorganic carbon (sDIC).

and salinity-normalized alkalinity (sALK) is roughly equivalent to sDIC over the first 5 lead years, while sea surface salinity plays a negligible role over all 10 lead years (Fig. 4c). Predictability in sDIC is mainly driven by the persistence of its anomalies, but is enhanced further by initializations (Supplementary Fig. 5). A budget analysis of DIC in the upper 150 m of the CCS suggests that variability in vertical and lateral DIC advection plays a leading role in setting the DIC inventory (Fig. 7), as evidenced by the high correlation between the advective flux and total tendency terms ($r = 0.9$). Source waters for the CCS exhibit substantial interannual to decadal variability and are mainly comprised of subarctic waters transported by the California Current (upper 200 m) and eastern tropical Pacific waters transported by the California Undercurrent (200–300 m), which propagate biogeochemical anomalies into the system[32,33]. Thus, the subsurface and basin-wide initializations of DIC—as well as predictability of meridional and vertical transport variability—are crucial factors in making skillful multiyear predictions of surface pH variability. In turn, enhanced observations or reanalysis of these fields would be necessary for operational forecasting of surface pH in the CCS.

## Discussion

While this study presents a very promising first result, there are some caveats worth noting. Simulations were run with a spatial resolution of ~100 km×100 km. In turn, we do not explicitly resolve the fine-scale coastal upwelling of corrosive waters (which occurs within roughly 30 km of the coastline in the CCS), but instead

simulate the combined effect of coastal and curl-driven upwelling in nearshore grid cells[27]. Our simulation also uses subgrid scale parameterizations to capture the important process of eddy-induced offshore flux of tracers in the CCS[34,35]. Despite the coarse resolution, alongshore winds, upwelling, air–sea $CO_2$ fluxes, surface $pCO_2$, and surface pH are well-represented in this configuration of CESM relative to observations[27,28]. However, the coarser grid resolution suppresses variability in surface pH. In turn, the annual surface pH anomalies being predicted are <0.01 units (Fig. 1b), but these relatively small anomalies are associated with large fluctuations in other environmentally relevant variables, such as the aragonite saturation state, which varies on the order of 0.1 units (Supplementary Fig. 6d). In spite of the relatively small target anomalies being predicted, CESM-DPLE forecast error (as measured by the NMAE) falls within the spread of the historical surface pH variability (Figs. 4b and 6). In this study, we only highlight predictability in annual averages of surface pH, since predictability at annual resolution is much higher than that of monthly resolution. However, we do find significant predictability of monthly surface pH anomalies over forecasts of persistence and external forcing through June of the upwelling season following initialization, and into April of the following upwelling season (Supplementary Fig. 7). We focus on assessing predictability in surface pH after removing the ocean acidification trend to highlight the role of initialization in engendering predictability. Our results are similar if we conduct the analysis on surface pH while retaining the ocean acidification signal (Supplementary Fig. 8). Lastly, in

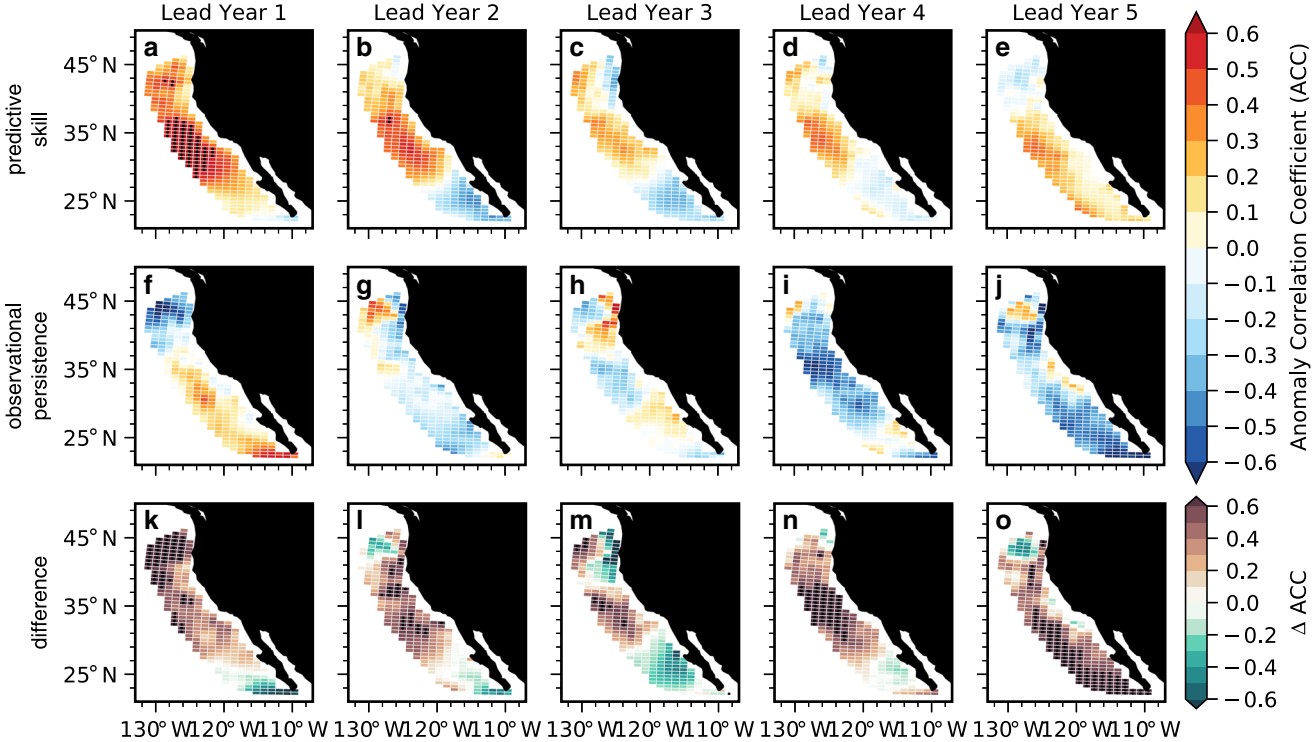

**Fig. 5 Predictive skill of surface pH anomalies. a–e** Community Earth System Model Decadal Prediction Large Ensemble (CESM-DPLE) initialized forecasts of detrended annual surface pH anomalies for lead years one through five correlated with the observational product over 1990–2005. **f–j** Persistence forecast for the observations for lead years one through five. Stippling in **a–j** denotes statistically significant correlations at the 95% level using a *t* test. An effective sample size is used in the *t* test to account for autocorrelation in the two time series being correlated. **k–o** Difference between the CESM-DPLE forecast anomaly correlation coefficients (ACCs) and observational persistence. Stippling indicates that the initialized prediction is statistically significant over the observational persistence forecast at the 95% level using a *z* test. Only positive ACCs and ΔACCs are stippled.

assessing predictive skill, we are challenged by the limited coverage of gridded surface pH observations. While the observational product used in this study spans 1990–2017, the observational data for atmospheric $CO_2$ used to force the reconstruction ended in 2005, after which point a smooth scenario-based projection was used (Supplementary Fig. 1a). This causes a drop-off in the ability of the reconstruction to replicate observed surface pH anomalies (Supplementary Fig. 1b). Thus, we only assess skill over the 1990–2005 period, limiting our degrees of freedom for statistical significance.

Our results demonstrate for the first time the potential for an initialized ESM to retrospectively predict surface pH multiple years in advance in a complex, sensitive, and economically important oceanic region. Although these forecasts cannot aid directly in the management of coastal fisheries at this spatial resolution, our results demonstrate the feasibility of making skillful surface pH predictions on multiannual to decadal timescales. Further, our work suggests that global initialized ESM forecasts can be used as boundary conditions to improve existing regional biogeochemical forecasting and to extend their lead times. Dynamically downscaled decadal forecasts with high-resolution regional models could go a long way toward improving fisheries management in sensitive coastal regions on interannual timescales. While our study highlights CESM-DPLE's ability to predict surface pH anomalies, other ocean acidification parameters—such as calcium carbonate saturation states—can be expected to be predictable, due to their common dependence on variability in dissolved $CO_2$. By detrending our simulated and observational products prior to analysis, we show that we have the potential to predict interannual variations in surface pH. As the ocean acidification signal dominates in this region over decadal timescales, multiyear predictions of surface pH variability

could aid in forecasting the acceleration or slowdown of ocean acidification in the CCS.

## Methods

**Model simulations.** The Community Earth System Model Decadal Prediction Large Ensemble[22] (CESM-DPLE) is based on CESM, version 1.1, and uses the same code base, component model configurations (Supplementary Table 2), and historical and projected radiative forcing as that used in its counterpart, the CESM Large Ensemble[24] (CESM-LE). This includes historical radiative forcing (with volcanic aerosols) through 2005 and projected radiative forcing (including greenhouse and short-lived gases and aerosols) from 2006 onward. The main difference between the two experiments is that CESM-DPLE is re-initialized annually to generate forecast ensembles (see next paragraph for details), while CESM-LE is only initialized once. We follow the convention of the decadal prediction community[21] and refer to the former as the initialized ensemble and the latter as the uninitialized ensemble. Because CESM-DPLE and CESM-LE have an identical code base and boundary conditions, the two ensembles can be compared directly to one another to isolate the relative influence of re-initialization and external forcing on hindcast predictability and skill.

CESM-DPLE was generated via full-field initialization each year on 1 November from 1954 to 2017, for a total of 64 initialization dates[21]. An ensemble of 40 forecast members was created by making Gaussian perturbations to the initial atmospheric temperature field (order $10^{-14}$ K) at each grid cell. Ensemble spread in all other fields and model components developed as a result of the spread in the atmospheric state. Each member was integrated forward from each initialization for 122 months, resulting in ~26,000 global fully coupled simulation years, costing roughly 50 million core hours to compute. The atmosphere and land components were initialized from the November 1st restart files of a single arbitrary member of CESM-LE (ensemble member 34)[36]. The atmosphere component is the Community Atmosphere Model, version 5 (CAM5) with a finite-volume dynamical core at nominal 1° resolution and 30 vertical levels[21,37]. Details on the land component can be found in Supplementary Table 2.

The ocean (including biogeochemistry) and sea ice model components in CESM-DPLE were re-initialized from the November 1st restart files of a forced ocean-sea ice reconstruction (referred to as the reconstruction; see the following paragraph for configuration details). The ocean biogeochemical model used in all CESM simulations in this study is the Biogeochemical Elemental Cycling (BEC)

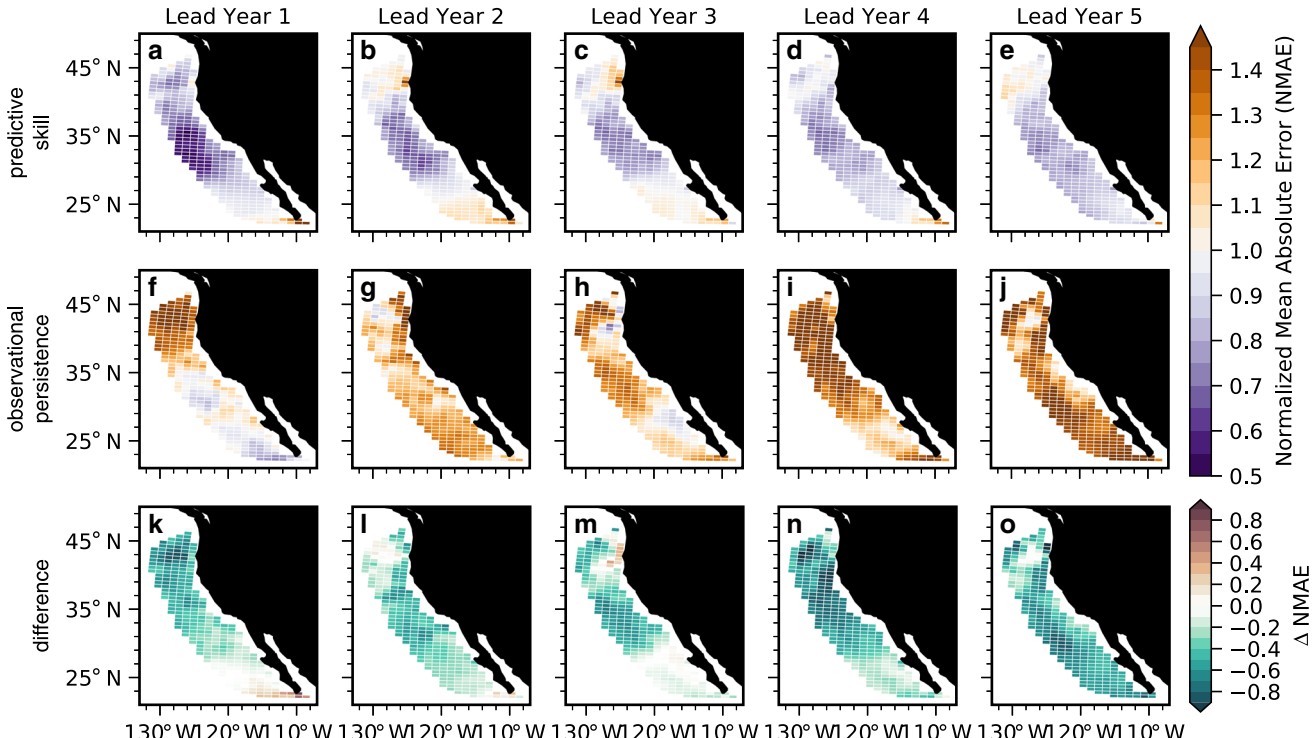

**Fig. 6 Normalized bias of surface pH anomaly forecasts. a–e** Normalized mean absolute error (NMAE) of Community Earth System Model Decadal Prediction Large Ensemble (CESM-DPLE) initialized forecasts of detrended annual surface pH anomalies for lead years one through five relative to the observational product over 1990–2005. **f–j** NMAE of a persistence forecast for the observations for lead years one through five. Purple colors (values below one) indicate that the forecast error is smaller than the interannual variability of observations; orange colors (values above one) indicate that the forecast error is larger than the interannual variability of observations. **k–o** Difference between the CESM-DPLE forecast and observational persistence NMAEs. Green colors indicate that the initialized forecasts have lower error than the persistence forecast.

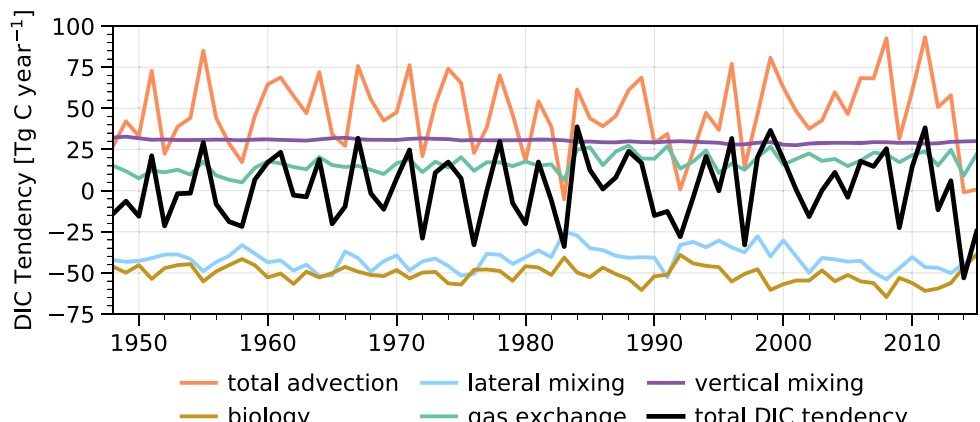

**Fig. 7 Dissolved inorganic carbon budget of the California Current.** Time series of the individual annual tendency terms of dissolved inorganic carbon (DIC) in the model reconstruction integrated over the California Current System laterally and to 150 m vertically (the approximate mean mixed layer depth in the model reconstruction). The colored lines show the individual terms, while the black line shows the total integrated DIC tendency.

model, which contains three phytoplankton functional types (diatoms, diazotrophs, and a small calcifying phytoplankton class), explicitly simulates seawater carbonate chemistry, and tracks the cycling of C, N, P, Fe, Si, and O[38,39]. Note that the ocean biogeochemistry and simulated atmospheric $CO_2$ concentration are diagnostic, such that there is no feedback onto the simulated physical climate[21]. Further details on drift adjustment and anomaly generation can be found in the following section.

The reconstruction simulation was run from 1948 to 2017 with active ocean (physics and biogeochemistry) and sea ice model components from CESM, version 1.1, with identical spatial resolutions as the fully coupled CESM-DPLE and CESM-LE (Supplementary Table 2). The ocean and sea ice components were forced by a modified version of the Coordinated Ocean-Ice Reference Experiment (CORE) with interannual forcing[40,41], which provides momentum, freshwater, and buoyancy fluxes between the air–sea and air–ice interfaces. CORE winds were used

globally, save for the tropical band (30S–30N), where NOAA Twentieth Century Reanalysis, version 2[42] winds (from 1948 to 2010) and adjusted Japanese 55-year Reanalysis Project[43] winds (through 2017) were used to correct a spurious trend in the zonal equatorial Pacific sea surface temperature (SST) gradient[21]. No direct assimilation of ocean or sea ice observations was used in the reconstruction; thus, any faithful reproduction of ocean and sea ice climatology or variability is due mainly to the atmospheric reanalysis that drives the simulation[21].

**Drift adjustment**. Initialized forecasts required drift adjustment due to the use of full-field initialization for the CESM-DPLE. To correct for this model drift, we followed the same procedure as in Yeager et al.[21]. Drift (i.e., lead-time dependent model climatology) was computed as the mean across ensemble members and start

dates, separately for each lead time range considered, where only those hindcasts that verify between 1964 and 2014 are included in the climatology. This drift was then subtracted at each grid cell from all forecasts to generate anomalies. Anomalies were computed for CESM-LE and the reconstruction by subtracting the mean over 1964–2014 at each grid cell. The same was done for the JMA observational product, but over the 1990–2005 period that the CESM-DPLE forecasts were verified against. A second-order fit was removed from all time series over their verification window.

**Observational product**. We compare initialized forecasts of surface pH to the Japanese Meteorological Agency (JMA) Ocean $CO_2$ Map product[25,26], which provides monthly estimates of surface pH from 1990 to 2017 over a 1° × 1° global grid. Here, we describe the key steps followed by the authors of the JMA product to derive their surface pH estimates. Surface pH was computed diagnostically with a carbonate system solver, using estimated surface alkalinity and $pCO_2$ as inputs. To compute gridded alkalinity, the ocean was divided into five regions, where empirical relationships were derived for in situ alkalinity as a function of sea surface height (SSH) and sea surface salinity[25] (SSS). Gridded observations of SSH and SSS (independent of the in situ observations) were then input into the empirical equations to derive gridded surface alkalinity. Gridded surface $pCO_2$ was computed through a multistep process. First, the ocean was divided into 44 regions and relationships between in situ $pCO_2$ and in situ SST, SSS, and Chl-$a$ were derived by multiple linear regressions in each region for one to three of the variables[26]. The gridded $pCO_2$ product was then derived by applying these functions to independent gridded observations of SST, SSS, and Chl-$a$. There are no uncertainty estimates available for the pH product, but the authors report a root mean square error (gridded estimate compared to in situ observations) of 10–20 μatm for $pCO_2$ in the northern hemisphere mid-latitudes and 8.1 μmol kg$^{-1}$ for surface alkalinity relative to the PACIFICA campaign[25,26]. Note that the global average JMA surface pH is within the uncertainty of the SOCAT-based estimate for all years (Supplementary Fig. 9). Further details on the datasets used in deriving their product can be found in Takatani et al.[25] and Iida et al.[26].

**Statistical analysis**. We use deterministic metrics to compare the ensemble mean retrospective forecasts to a persistence forecast, and in some cases, the uninitialized CESM-LE ensemble mean forecast. A comparison of the initialized forecast to the persistence forecast shows the utility of our initialized forecasting system over a simple, low-cost forecasting method; a comparison of the initialized forecast to the uninitialized forecast shows the utility of initializations (rather than external forcing) in lending predictability to the variable of interest. The persistence forecast assumes that anomalies from each initialization year persist into all following lead years (or months)[44]. The uninitialized forecast compares the CESM-LE ensemble mean anomalies to the verification data (model reconstruction or observations) over the same window as the initialized forecasting system[21]. Unless otherwise noted, forecasts are analyzed at annual resolution. This corresponds to the January–December average following the 1 November initialization. In turn, lead year 1 truly covers lead months 3–14. When considering monthly predictions, lead month one corresponds to the 1–30 November average following initialization.

We compute the anomaly correlation coefficient (ACC) via a Pearson product-moment correlation to quantify the linear association between predicted and target anomalies (where the target is either the model reconstruction or observational product). If the predictions perfectly match the sign and phase of the anomalies, the ACC has a maximum value of 1. If they are exactly out of phase, it has a minimum value of −1. The ACC is a function of lead time[10,45]:

$$\mathrm{ACC}(\tau) = \frac{\left(\sum_{\alpha=1}^{N} \left(F\prime_{\alpha}(\tau) \times O\prime_{\alpha+\tau}\right)\right)}{\sqrt{\sum_{\alpha=1}^{N} F\prime_{\alpha}(\tau)^2 \sum_{\alpha=1}^{N} O\prime_{\alpha+\tau}^2}}$$

Where $F\prime$ is the forecast anomaly, $O\prime$ is the verification field anomaly, and the ACC is calculated over the initializations spanning 1954–2017 ($N = 64$) relative to the reconstruction and CESM-LE, and over initializations covering 1990–2005 ($N = 16$) relative to the JMA observational product. We quantify statistical significance in the ACC using a $t$ test at the 95% confidence level with the null hypothesis that the two time series being compared are uncorrelated. We follow the methodology of Bretherton et al.[46], using the effective sample size in $t$ tests to account for autocorrelation in the two time series being correlated:

$$N_{\mathrm{eff}} = N\left(\frac{1 - \rho_1\rho_2}{1 + \rho_1\rho_2}\right)$$

Where $N$ is the true sample size and $\rho_1$ and $\rho_2$ are the lag 1 autocorrelation coefficients for the forecast and verification data. We assess statistical significance between two ACCs (e.g., between that of the initialized forecast and a simple persistence forecast for the same lead time) using a $z$ test at the 95% confidence level with the null hypothesis that the two correlation coefficients are not different.

To quantify the magnitude of forecast error, or the accuracy in our forecasts, we use the normalized mean absolute error[45] (NMAE), which is the MAE normalized by the interannual standard deviation of the verification data. The NMAE is 0 for perfect forecasts, <1 when the forecast error falls within the variability of the

verification data, and increases as the forecast error surpasses the variability of the verification data. MAE is used instead of bias metrics such as the root mean square error (RMSE), as it is a more accurate assessment of bias in climate simulations[47].

$$\mathrm{NMAE}(\tau) = \frac{1}{N}\sum_{\alpha=1}^{N} \frac{|F\prime_{\alpha}(\tau) - O\prime_{\alpha+\tau}|}{\sigma_{O\prime}(\tau)}$$

Where $N$ is the number of initializations and $\sigma_{O\prime}$ is the standard deviation of the verification data over the verification window.

**Linear decompositions**. We follow Lovenduski et al.[17] to convert predictability in pH driver variables (SST, SSS, sDIC, and sALK) to common pH units:

$$r_x \cdot \frac{\mathrm{dpH}}{\mathrm{d}x} \cdot \sigma_x$$

Where $r_x$ is the ACC between anomalies in driver variable $x$ and target anomalies, $\frac{\mathrm{dpH}}{\mathrm{d}x}$ is the linear sensitivity of pH to the driver variable, and $\sigma_x$ is the standard deviation of driver variable anomalies in the reconstruction.

We use a linear Taylor expansion to quantify the relative contribution of variability in environmental drivers to total surface pH variability in the CCS[28,48]:

$$\mathrm{pH}\prime = \frac{\mathrm{dpH}}{\mathrm{d}T}T\prime + \frac{\mathrm{dpH}}{\mathrm{d}S}S\prime + \frac{\mathrm{dpH}}{\mathrm{dDIC}}\mathrm{sDIC}\prime + \frac{\mathrm{dpH}}{\mathrm{dALK}}\mathrm{sALK}\prime + \mathrm{residual}$$

Where primes denote annual average anomalies after removing a second-order polynomial fit, and $\frac{\mathrm{dpH}}{\mathrm{d}x}$ the linear sensitivity of pH to the driver variable $x$. Residual variability is due to freshwater dilution effects, higher-order terms excluded in the linear expansion, and cross-derivative terms[28]. Sensitivities were computed using the carbonate system solver, CO2SYS. For example, $\frac{\mathrm{dpH}}{\mathrm{d}T}$ was computed by varying SST by its seasonal range in the CCS in the model reconstruction while holding DIC, alkalinity, and salinity constant at their mean values in the CCS. A linear slope was then fit to the resulting change in surface pH over this range.

## Data availability
Output from the Community Earth System Model Decadal Prediction Large Ensemble (CESM-DPLE) and CESM model reconstruction can be downloaded at [https://www.earthsystemgrid.org/dataset/ucar.cgd.ccsm4.CESM1-CAM5-DP.html]. Output from the CESM Large Ensemble (CESM-LE) can be downloaded at [http://www.cesm.ucar.edu/projects/community-projects/LENS/data-sets.html]. The JMA Ocean $CO_2$ map product can be downloaded online at [https://www.data.jma.go.jp/gmd/kaiyou/english/co2_flux/co2_flux_data_en.html].

## Code availability
Analysis was performed using climpred, an open source python package developed by Riley Brady and Aaron Spring (MPI) for analyzing initialized forecast models. Documentation is available at https://climpred.readthedocs.io. The code used to create all figures in this study is available on Github [https://github.com/bradyrx/california_current_ph_forecasts].

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

## Acknowledgements

The CESM project is supported primarily by the National Science Foundation (NSF). This material is based upon work supported by the National Center for Atmospheric Research, which is a major facility sponsored by the NSF under Cooperative Agreement No. 1852977. Computing and data storage resources, including the Cheyenne supercomputer (doi: 10.5065/D6RX99HX), were provided by the Computational and Information Systems Laboratory (CISL) at NCAR. The Department of Energy's Computational Science Graduate Fellowship supported RXB throughout this study (DE- FG02-97ER25308). N.S.L. and R.X. B. are grateful for support from the NSF (OCE-1752724). R.X.B. acknowledges Aaron Spring for his contributions to analysis through collaborative development of the climpred package as well as Samantha Siedlecki, Michael Jacox, and Michael Alexander for suggestions during the analysis phase of the project.

## Author contributions

R.X.B. and N.S.L. designed the study. R.X.B. analyzed the data, prepared figures and tables, and wrote the paper. S.G.Y. and K.L. coordinated and ran CESM-DPLE and FOSI simulations. N.S.L., S.G.Y., M.C.L., and K.L. provided invaluable feedback throughout the study and reviewed the manuscript.

## Competing interests

The authors declare no competing interests.
