## [Peer Review File · Nature Communications]

Reviewers' comments:

Reviewer #1 (Remarks to the Author):

Review of 'Skillful multiyear predictions of ocean acidification in the California Current System' by Brady et al.

In this manuscript, Brady et al. assess the capability of an Earth system model (here CESM) to predict observed year-to-year variations in surface pH in the California Current System. The authors show that CESM can predict those variations up to 5 years in advance. The analyses conducted here are well designed and well described. Besides, I much appreciate that the authors use an accurate wording when defining predictive skill and potential predictability. In terms of novelty, this study puts together a new observational dataset spanning from 1990 to 2017 and a new large ensemble dataset of model predictions. I think this paper needs some clarification that have to be addressed first, and which prevent me of accepting this paper in its present form.

First of all, the paper looks to be fitted for a Letter format whereas 'Nature Communications' offers more space to detail some parts of your work and further details on model properties and caveats. Given that space, I would suggest the authors to dive a bit further into the model evaluation because it represents one of the major caveats of this study. Coarse resolution ocean models are known to poorly represent coastal ocean dynamics and upwelling and have difficulties to replicate coastal ecosystem dynamics. Having a good predictability doesn't mean that the model processes are accurate. That's why I think the authors need to support their finding with a deeper evaluation of the domain of interest. In particular, the evaluation of ocean properties such as ocean circulation, sea-level anomalies in the California Current System would help to say (and confirm) that the major feature of ocean physics in the CCS are accurately reproduced by the model. This analysis can help to flesh out the discussion on the processes behind the multi-year predictability. I think the other key properties of coastal system need to be discussed as well. For example, there are several other database that may help to discuss the key carbon-related properties (such as Laruelle et al. 2017, Laruelle et al. 2018).

Second, a deeper evaluation of the dataset produce in this work would be also helpful. How does this dataset compare to SOCAT-based mean-state pH estimate (http://marine.copernicus.eu/services-portfolio/access-to-products/?option=com_csw&view=details&product_id=GLOBAL_OMI_HEALTH_carbon_ph_area_averaged) ?

It would be also helpful to further discuss the main characteristics of the ocean pH (inferred from the observational dataproduct) in the CCS ? For instance:

- Does the year-to-year variations in surface pH are larger than the long-term trends ?
- How does these variations correlate with environmental drivers ?
- What is the decorrelation time-scales of pH anomalies ?

Investigating the use of other database for ocean pH (such as SOCAT) may help to better constraint the statistics use to assess the predictability of the model.

Minor comments:

L2: please detail why the upwelled water are corrosive.

L5: I would suggest to remove novel. Use 'advance' instead

L8: given accurate number of year instead of "multiple"

L46-47 please provide further details

L54: most powerful 'approach' not a tool

L93: it is unclear what do you mean with second-order ? polynomial fit ?

Figure 3 and in the related text: please detail somewhere the R^* corresponding to the significance limits for your prediction.

You mentioned that you use a t-test. Pending on the above suggestion, it maybe good to use the

Bretherthon et al. (1999) to compute the degree of freedom use for your t-test.

L123: What does 'virtually' mean ?

L126: ΔACC = difference between ACCs ?

Reviewer #2 (Remarks to the Author):

The paper describes an analysis the predicatability of surface pH in the California Current System using Earth System Models. The paper reports that ESMs re-initialized in November of each year (CESM-DPLE) have statistically improved, several year predictive skill of modeled and estimated surface pH based on empirical multivariate statistical relationships than either non-re-initialized simulations (CESM-LE) or persistence based on what the authors refer to as the reconstruction. The authors identify that predictive skill stems from advection of DIC anomalies. The analysis is sound, the paper is quite well-written, and the results are interesting. I recommend publication with minor revisions.

My main comments for the paper are as follows.

1) It's surprisingly difficult to understand the different simulations. All models are initialized at some point in time, so calling the CESM-DPLE runs "initialized" is not helpful. Perhaps re-initialized is more accurate?

More importantly, in the Methods section where the runs are described, considerably more care should be taken in explaining the different runs. While I read this section several times, I still don't think I understand. It is my understanding from the text that the CESM-DPLE runs are initialized as follows.

The atmospheric temperature portion is created somehow by round-off perturbations. What does that mean? $1e-6$ relative changes in values, randomly at all points, or with some assumed spatial or temporal decorrelation? Are these $1e-16$ changes? These changes seem negligible to me, but perhaps they are significant. Also confusing: on line 205, it mentions the round-off perturbations to temperature, but then shifts to the ocean-sea ice model and then shifts back to the atmospheric model on line 209. The text says the atmosphere is initialized from a single ensemble member. Is that member randomly chosen? Would the the ensemble mean be better? It would be helpful to describe the atmospheric component in full before introducing the ocean-sea ice component (line 207).

The ocean-sea ice values in re-initialization comes from the reconstruction. Seems ok, except that in Figure 1B, the blue dots (forecast means) don't sit on top of the black line (reconstruction). Why is that?

I assume that the BEC initialization also comes from the reconstruction, though this initialization is not stated anywhere.

Please follow the spirit of this point by rewriting or significantly improving the Methods section so that the distinction between CESM-LE, CESM-DPLE, reconstruction, and persistence are crystal clear.

2) Looking at Figure 1, it seems plausible that the blue lines are better predictors of the red lines than persistence of the reconstruction and than the un-re-initialized run (CESM-LE), but this is not shown. Can you add a line corresponding to the CESM-LE run to Figure 1 without over-cluttering it?

3) Is there any estimate of the uncertainty in the observational estimates? The agreement between the area weighted reconstruction and the observational estimates suggests that the variability is real, but there's probably considerable uncertainty in the red-line (and individual points that go into Fig 4

calculations) that reduce significance. It's possible that you've already accounted for this, but I don't see discussion of it.

4) There is probably not space for this, but since you make a prediction for 2020 being more basic at the coast and more acidic offshore, can you explain that signal more? At first I was surprised, simply assuming that the coastal anomaly should always be more acidic than the area-weighted mean. But your later results suggest that my first interpretation was wrong, that it's the subsurface DIC anomaly that is anomalously low prior to upwelling, causing a reduced near coast signal in 2020? An alternative might be reduced alongshore winds in 2020 or a change in the timing of the onset of coastal upwelling? Given your prediction, it would be nice if you can trace it back to the driver within the model. The obvious place to do this would be on the section discussing the significance of sDIC.

Reviewer #3 (Remarks to the Author):

The study by Brady and Coauthors discusses a new application of an ocean forecast system to predict the progression of ocean acidification (specifically, surface pH) in the California Current System (CCS). They show that the forecast is more skillful than naive "persistence" forecasts, and is able to capture some of the variability in acidification several year in advance.

Ocean acidification is one of the main anthropogenic stressors threatening marine ecosystems, and has received increasing attention over the past decade or so. Ocean biogeochemistry forecasting is still at the early stages, with several interesting results recently published (e.g. Park et al., 2019, Science). This paper merges the two topics into a coherent framework, and perhaps represent the first example, to my knowledge, of acidification forecasting on multi-year timescales. (Skillful forecasts of other carbon cycle processes, e.g. carbon uptake, have been presented in other studies.)

The paper is well written, and is clearly the result of a significant amount of work, in particular an impressive amount of of simulations with a state-of-art Earth System Model (ESM) ensemble "machinery". As such, I think the paper is definitely publishable. However, I have some concerns that make me question publication in Nature Communication. These are mostly related to the ability of the ESM to provide credible simulation (and forecast) of the patterns of acidification at the local scales that are relevant for management of marine resources. Thus, I think that the potential of the forecast described in the paper to provide predictions relevant to management is inflated. My impression is that the paper could be more suitable to a field-specific journal, as a first step towards useful regional predictions of acidification or biogeochemistry.

Main Comments:

- The approach is based on an ESM with a horizontal resolution of 100km. This is insufficient to capture the spatial gradients of acidification (pH, saturation state) that naturally occurs along the CCS, both cross-shore and alongshore. These are the relevant gradients for many important fisheries in the CCS, in particular the ones that are more susceptible to acidification (e.g. Dungeness crabs, shellfish), which are very coastal and confined to the shelf. I have a hard time considering a 100-km resolution forecast relevant in an oceanographic region dominated by circulation and gradients at scales of order of few 10s of km, i.e. the coastal upwelling band, the shelf, and the alongshore heterogeneity driven by topography. There has been a fairly long list of papers showing how capturing these scales in the ocean (and even atmosphere, as forcing) is important to reproduce realistic physical and biogeochemical gradients in upwelling systems. Some of these studies have shown that ocean model resolution really matters in the CCS, with coarse models unable to represent realistic spatial patterns in physical and C cycle variables. For example, Fiechter et al., 2014, Global Biogeochemical Cycles, showed that even models at 30km resolution are dramatically biased in their representation of surface pCO₂ (closely related to pH), compared to models at resolution less than 10km. Other work has shown that factors like the shape of winds at scales of ~10km control the strength of coastal vs.

offshore upwelling, determining the redistribution of nutrient- and DIC-rich waters (Renault et al., 2018, Nature Geosciences). Interactions of winds and currents with coastal topographic features at scales of 10s-100km also shape productivity, surface chlorophyll, and presumably C-system parameters and pH (Fiechter, et al., 2018, Global Biogeochemical Cycles). Similarly, the coastal undercurrent that carries DIC-rich waters from the tropics just below the surface along the coast, and serves as a water reservoir for coastal upwelling, is not represented at scales of 100km. In summary, the tools utilized in the paper seem more appropriate for a large-scale analysis, rather than a coastal analysis that could be relevant for marine resource management. The Authors are of course aware of these shortcomings, and discuss them in the manuscript; however, they are important, and they limit the ability of the forecast to provide information at the relevant scales.

- Perhaps as a consequence of the coarse resolution, the acidification signals that the paper show to be predictable on multi-year time scales are surprisingly small. As such, it is not clear that they would have a significant influence on the ecosystem, or that they would be useful for management decisions. For example, Fig. 1b-c indicates ability of predicting fluctuations on the order of 0.003 pH units. This are very small variations (roughly corresponding to a change in H⁺ concentration of less than 1%!). My sense is that the type of pH variation that could drive significant ecological changes (e.g. changing the saturation state of carbonates by a significant amount, e.g. on the order of 0.1 saturation units) would need to be at least an order of magnitude larger, if not more. The paper does not discuss in detail the potential importance for ecosystem, and hence for management, of the type of fluctuations that are shown to be predictable. This should be addressed to make a strong case for the usefulness of the multi-year pH forecast for applications. The Authors should strive to convince the readers that the predicted perturbations actually matter for ecosystem and management, as claimed in the introduction.

- The main signal in interannual variations of pH and C cycle tracer is a forced signal driven by the atmospheric pCO₂ increase (as discussed for example in Gruber et al., 2012, Science). This is a predictable, major signal, and yet is by construction discarded by the Authors. I understand that predicting fluctuations driven by internal dynamics is a much more challenging (and dynamically interesting) problem that requires a major ESM-based forecast machinery. But nonetheless, removing this large, multi-annual signal forces the Authors to focus on small residual perturbations (Fig. 1b-c) that are probably not as impactful.

- Additionally, member-to-member anomalies (thin lines in Fig. 1b) are substantially larger than the model ensemble mean anomaly (as are reconstructions and observations), suggesting that it is really poorly- or non-predictable fluctuations that could have a greater impact on pH, and possibly on ecosystem. Larger, shorter fluctuations could be driven by variability that has a weak "memory", e.g. winds fluctuations driving anomalous advection or upwelling. The existence of larger, unpredictable fluctuations also reduces the direct utility of the multi-annual forecast shown in the paper. That said, even these individual-member fluctuations appear quite minor when compared to the multi-year progression of acidification in Fig. 1a, which is driven by the atmospheric pCO₂ increase.

- On a related note, I wonder to what extent potentially-damaging acidification events occur at the spatiotemporal scale of upwelling events, which are far less predictable than the broader-scale, more persistent conditions that are addressed in this study.

- Finally, the paper feels a bit short on broader implications, and the case for marine resource managers or other stakeholders to use this type of multi-annual forecasts is weak (e.g. in the introduction and discussion). As a comparison, the paper by Park et al. 2019, Science, was able to connect predictability in SST and Chl to fishery catch predictions (albeit on shorter timescales), which has more obvious direct implications.

- That said, my sense is that the work presented by Brady et al. is significant, and could be indeed useful in connection to resource management, for example to help downscaled biogeochemical predictions. For example, it could be used to initialize and drive regional biogeochemical forecast systems, of which a few are being developed along the US West Coast (e.g. Siedlecki et al., 2016, Sci. Rep., etc.). But this is beyond what is discussed in the manuscript.

Other comments:

- Line 28: I am not sure pteropods, coccolithophores, and shellfish are “keystone” species in the CCS. This statement should be better supported. Likewise, the most valuable commercial fisheries in California today are Dungeness crab and market squid, and I am not sure how either connect to the species listed.

- Lines 33-34. Shellfisheries tend to be fairly coastal, and affected by small-scale (e.g. coastal upwelling) processes. Also, I am not sure buffering fishery sites with sodium carbonate is something that managers realistically consider, given costs, scale, effectiveness, and legal implications.

- Lines 42-50. Maybe more details on what management approaches have indeed included information on multi-annual to decadal timescales would help the argument. To my knowledge, most fishery assessments are based on annual projections. Also, I am not aware of fishery management that includes ocean biogeochemistry variables. Examples would help.

- Lines 130-136. This is a cool result, but again the forecast signal has a pretty small magnitude.

- Fig. 4. Why is the coastal band omitted in these figures? Is it missing from the observational database? Unfortunately, important marine resources are distributed along this band.

- Line 150 and Fig. S4. This is also a cool analysis.

- Lines 231-246. I am confused by this section. Is this a description of the JMA product, or was the calculation re-done as part of this paper? This should be clarified.

- Line 280. “Increase to infinity”. This seems extreme. To go to infinity, either the forecast or the verification need to go to infinity, which seems unphysical.

Response to Reviewers

In response to the helpful comments from our three referees, we have substantially revised the originally submitted manuscript. In response to Reviewer #3's concerns about the 100 km resolution of the model, we have expanded analyses and have added text to the abstract, results, and discussion. We thoroughly evaluate the model in the results (rather than in the supplemental) to illustrate its ability to capture the mean state and variability of surface pH and other carbonate chemistry variables in the California Current system. We also mention the caveats of using this coarse simulation and reduce language about applying our results directly to regional fisheries. In response to their concerns about the magnitude of forecasted surface pH anomalies, we added a supplemental figure (Fig. S7) and text to the discussion, showing that smaller fluctuations in surface pH are associated with large fluctuations in other environmentally important variables. We also changed our forecast bias metric from mean absolute error (MAE) to normalized MAE (NMAE), which shows that our forecasts are accurate enough to fall within the historical variability of the system (Figs. 4, 6). In response to Reviewers #1 and #2, we substantially rewrote the methods section to clarify the experimental setup of our simulations. We also added multiple analyses to evaluate our model reconstruction against observations (Figs. S3 and S4) and verified the JMA observational product against an independent SOCAT-based product (Fig. S9). Lastly, we adapted all of our significance testing to account for autocorrelation in the two time series being correlated, following Bretherton et al. 1999 (Figs. 2, 3, 4, 5, S1, S5, S6, S8).

Reviewer #1 (Remarks to the Author):

Review of 'Skillful multiyear predictions of ocean acidification in the California Current System' by Brady et al.

In this manuscript, Brady et al. assess the capability of an Earth system model (here CESM) to predict observed year-to-year variations in surface pH in the California Current System. The authors show that CESM can predict those variations up to 5 years in advance. The analyses conducted here are well designed and well described. Besides, I much appreciate that the authors use an accurate wording when defining predictive skill and potential predictability. In terms of novelty, this study puts together a new observational dataset spanning from 1990 to 2017 and a new large ensemble dataset of model predictions. I think this paper needs some clarification that have to be addressed first, and which prevent me of accepting this paper in its present form.

We thank the anonymous reviewer for their comments that helped to substantially improve this manuscript.

First of all, the paper looks to be fitted for a Letter format whereas 'Nature Communications' offers more space to detail some parts of your work and further details on model properties and caveats.

Thank you for mentioning this. We have moved "Model Evaluations" to the main text and added additional analyses, discussion, and figures throughout the manuscript to take advantage of this extra space.

Given that space, I would suggest the authors to dive a bit further into the model evaluation because it represents one of the major caveats of this study. Coarse resolution ocean models are known to poorly represent coastal ocean dynamics and upwelling and have difficulties to replicate coastal ecosystem dynamics. Having a good predictability doesn't mean that the model processes are accurate. That's why I think the authors need to support their finding with a deeper evaluation of the domain of interest. In particular, the evaluation of ocean properties such as ocean circulation, sea-level anomalies in the California Current System would help to say (and confirm) that the major feature of ocean physics in the CCS are accurately reproduced by the model. This analysis can help to flesh out the discussion on the processes behind the multi-year predictability.

We have moved the “Model Evaluation” portion of the supplemental into the main text, as we agree that one of the major caveats of the study is the coarse model resolution. We have also added the following lines of text to “Model Evaluation” to summarize the lead author's efforts to validate the California Current System circulation and carbon in previous studies:

“Previous evaluations of the physical circulation and carbonate chemistry in the version of CESM used for CESM-DPLE suggest that, despite the relatively coarse 1° x 1° model grid, CESM provides a good fit to observational climatologies of alongshore wind stress, surface pCO₂, and air–sea CO₂ fluxes in the CCS^{1,2}. Modeled alongshore wind stress—the primary driver of coastal upwelling—closely matches the magnitude and seasonality of observations, with peak upwelling-favorable conditions spanning April to September¹. The large-scale spatial structure of air–sea CO₂ fluxes in the model exhibits poleward CO₂ uptake and equatorward CO₂ outgassing, matching that of modern observationally based estimates^{2,3}. Importantly, we note that CESM cannot capture the nearshore outgassing of CO₂ associated with the coastal upwelling of carbon-enriched waters that occurs on a scale smaller than the resolution of the model grid^{2,4}. The modeled monthly climatology of area-weighted surface ocean pCO₂ in the CCS closely resembles that of the observationally based estimate, due to the model's proper simulation of the magnitude and phasing of thermal (solubility-driven) and non-thermal (circulation- and biology-driven) pCO₂ effects^{2,3}.”

I think the other key properties of coastal system need to be discussed as well. For example, there are several other database that may help to discuss the key carbon-related properties (such as Laruelle et al. 2017, Laruelle et al. 2018).

While we agree that it would be beneficial to validate the coastal area more rigorously, we don't feel that it is appropriate to compare our 1° x 1° degree resolution to the Laruelle solutions, which resolve the strong nearshore outgassing of CO₂ due to the upwelling of carbon-enriched waters. We don't expect to resolve this phenomenon with our version of CESM, since this occurs at scales smaller than the grid cell resolution. As a result, we focused on the LME as a whole. We've added the following sentence to “Model Evaluation” to clarify this point:

“Importantly, we note that CESM cannot capture the nearshore outgassing of CO₂ associated with the coastal upwelling of carbon-enriched waters that occurs on a scale smaller than the resolution of the model grid^{2,4}.”

Second, a deeper evaluation of the dataset produce in this work would be also helpful. How does this dataset compare to SOCAT-based mean-state pH estimate (http://marine.copernicus.eu/services-portfolio/access-to-products/?option=com_w&view=details&product_id=GLOBAL_OMI_HEALTH_carbon_ph_area_averaged) ?

Thank you for this suggestion. We compared the area-weighted mean of the JMA observational product to the SOCAT-based mean-state pH estimate. See the below figure, which has been added to the supplemental and is referenced in the Methods section where we introduce the JMA product:

“Note that the global average JMA surface pH is within the uncertainty of the SOCAT-based estimate for all years (Fig. S9).”

Fig S9. *Validation of JMA observational product against the SOCAT-based product. Global mean surface pH from the SOCAT-based product (black) and JMA observational product (red) used in this study. Gray shading denotes uncertainty bounds provided with the SOCAT-based product.*

It would be also helpful to further discuss the main characteristics of the ocean pH (inferred from the observational data product) in the CCS ? For instance:
- Does the year-to-year variations in surface pH are larger than the long-term trends ?

We have added the following lines to the model evaluation to address this:

“Over the area-weighted CCS (Fig. 2), the reconstruction simulates a linear change in pH of -0.026 over the 1990–2005 period, compared to the observational product’s linear

change of -0.029 (Fig. 2A). Both the reconstruction and observational product exhibit an interannual standard deviation of 0.003 in surface pH. Thus, the interannual variability in both the model and observations is between 1.5 to 2 times greater than the ocean acidification trend over the course of one year.”

Fig 2. Area-weighted temporal evaluation of surface pH in the model reconstruction. (A) Monthly surface pH in the California Current over 1990–2005 for the model reconstruction (black) and observational product (red). (B) As in (A), but for anomalies after removing a second-order polynomial fit and the seasonal cycle. The correlation coefficient between the observational product and model reconstruction is shown in the bottom left of (B). (C) As in the other panels, but for the mean monthly seasonal cycle over 1990–2005.

- How does these variations correlate with environmental drivers ?

We added a supplemental figure for this analysis and added the following text to “Model Evaluation”:

“We identify the drivers of reconstructed surface pH variability in the CCS by estimating the contributions from variations in salinity, alkalinity, SST, and dissolved inorganic carbon (DIC; see methods). The two major terms driving variability in surface pH are DIC and SST, whose standard deviation is approximately three times that of surface pH (Fig. S4). These two terms exhibit low-frequency variability and are significantly correlated with modes of variability such as the Pacific Decadal Oscillation (PDO) and El Niño–Southern Oscillation (ENSO). The linear correlation coefficient between DIC and SST residuals and the PDO is 0.66 and 0.73, and ENSO is 0.52 and 0.64, respectively (Table S1). Since surface pH is the small residual of many variables, it has a correlation coefficient of nearly zero with both modes of variability (Table S1).”

Fig S4. Influence of variability in environmental variables on surface pH variability in the California Current. (A) Interannual variability in surface pH due to individual environmental variables (see methods). Colors correspond to the color of each variable in (B). The dashed black line depicts the sum of all terms, or the total surface pH anomaly. (B) Magnitude of surface pH variability due to each environmental variable, quantified by the standard deviation from (A).

Table S1. Correlations between surface pH, environmental variables, and major modes of Pacific Ocean climate variability. Linear correlations are computed between annual averages of variables with a second-order polynomial fit removed. All variables except for surface pH represent the linear response of surface pH to that variable (see methods). Asterisks represent statistical significance per Bretherton et al. 1999⁵ with $\alpha = 0.05$.

Variable	Pacific Decadal Oscillation (PDO)	El Niño–Southern Oscillation (ENSO)
surface pH	-0.03	-0.07
dissolved inorganic carbon ⁺	0.66*	0.52*
temperature ⁺	-0.73*	-0.64*
alkalinity	-0.22	-0.04
salinity ⁺	-0.20	-0.14
residual	0.43	0.24

⁺An increase in these variables causes increased acidity, or a decrease in pH. Thus, a negative correlation coefficient represents a reduction in pH, but increase in the environmental variable.

- What is the decorrelation time-scales of pH anomalies ?

We addressed this in the “Model Evaluation” section of the text and added a supplemental figure:

“Surface pH anomalies in both the reconstruction and observational product exhibit a decorrelation time scale of four months (Fig. S3).”

Fig S3. *Autocorrelation function for surface pH in the California Current System.* Autocorrelation function computed for monthly anomalies of surface pH in the California Current over 1990–2005, after removing a second-order polynomial fit and the monthly climatology. The black dashed line represents the e-folding level ($1/e$), which defines the boundary for the decorrelation time scale.

Investigating the use of other database for ocean pH (such as SOCAT) may help to better constraint the statistics use to assess the predictability of the model.

Thank you for all of the above suggestions. We feel that this has substantially improved the model evaluation portion of our manuscript.

Minor comments:

L2: please detail why the upwelled water are corrosive.

We have updated the sentence to clarify this point:

“The California Current System (CCS) sustains economically valuable fisheries and is particularly vulnerable to ocean acidification, due to its natural upwelling of carbon-enriched waters that generate corrosive conditions for local ecosystems.”

L5: I would suggest to remove novel. Use ‘advance’ instead

We have opted to continue to use the word “novel” since this ensemble configuration is quite new to models including ocean biogeochemistry.

L8: given accurate number of year instead of “multiple”

We updated the abstract to reflect this:

“We show that the forecast system skillfully predicts observed surface pH variations a year in advance over a naïve forecasting method, with the potential for skillful prediction up to five years in advance.”

L46-47 please provide further details

We have rephrased this portion of the manuscript, per suggestions from Reviewer 3:

“Some level of skill can be provided by persisting anomalies from year to year in the system⁶. These so-called persistence forecasts are commonly used as a reference to put initialized skill into context and work at lead times commensurate with the decorrelation timescales of the system⁶⁻⁹.”

L54: most powerful ‘approach’ not a tool

We have changed “tool” to “approach” here.

L93: it is unclear what do you mean with second-order ? polynomial fit ?

We replaced “second-order trend” with “second-order polynomial fit” here and elsewhere in the manuscript.

Figure 3 and in the related text: please detail somewhere the R^* corresponding to the significance limits for your prediction.

We have added to the (now Figure 4) caption a description of the R^* range, and have added how this was computed to the methods:

“The critical value required for a statistically significant correlation ranges from 0.26 to 0.32 across leads, as computed by inverting the t statistic formula.”

You mentioned that you use a t-test. Pending on the above suggestion, it maybe good to use the Bretherthon et al. (1999) to compute the degree of freedom use for your t-test.

Thank you for this suggestion. We have remade Figures 2, 3, 4, S3, and S6 from the original manuscript to compute statistical significance using the effective sample size per Bretherthon et al. (1999). We have also added to the methodology a description of this version of the t test:

“We follow the methodology of Bretherton et al. (1999)⁵, using the effective sample size in t tests to account for autocorrelation in the two time series being correlated:

$$N_{eff} = N \left(\frac{1 - \rho_1 \rho_2}{1 + \rho_1 \rho_2} \right)$$

Where N is the true sample size and ρ_1 and ρ_2 are the lag 1 autocorrelation coefficients for the forecast and verification data.”

L123: What does ‘virtually’ mean ?

We updated this to reflect the fact that none of these grid cell correlations are statistically significant (with the updated Bretherton et al. p-values):

“Note, however, that none of these correlations are statistically significant at the 95% level.”

L126: Δ ACC = difference between ACCs ?

We added the following after Δ ACC:

“(the difference between ACCs for the initialized ensemble and observational product)”

Reviewer #2 (Remarks to the Author):

The paper describes an analysis the predictability of surface pH in the California Current System using Earth System Models. The paper reports that ESMs re-initialized in November of each year (CESM-DPLE) have statistically improved, several year predictive skill of modeled and estimated surface pH based on empirical multivariate statistical relationships than either non-re-initialized simulations (CESM-LE) or persistence based on what the authors refer to as the reconstruction. The authors identify that predictive skill stems from advection of DIC anomalies. The analysis is sound, the paper is quite well-written, and the results are interesting. I recommend publication with minor revisions.

We thank the anonymous reviewer for their comments that helped to substantially improve our manuscript.

My main comments for the paper are as follows.

1) It's surprisingly difficult to understand the different simulations. All models are initialized at some point in time, so calling the CESM-DPLE runs "initialized" is not helpful. Perhaps re-initialized is more accurate?

We use the term “initialized” for the CESM-DPLE (annual re-initializations with perturbations) and “uninitialized” for the CESM-LE (singular initialization with perturbations) as it is the standard in the field¹⁰. Although we do agree that this language is confusing, since as you mention, all models are initialized at some point in time. We decided to keep the “initialized” and “uninitialized” terminology for the main text but spent considerably more time introducing these in the methods (and used the phrase “re-initialized” in some places).

We removed “uninitialized” when first introducing CESM-LE in the methods (L196), changed “initialization” to “re-initialization” in L201, L207, L209, and added the following sentences for clarification in the first paragraph of the methods:

“The main difference between the two experiments is that CESM-DPLE is re-initialized annually to generate forecast ensembles (see next paragraph for details), while CESM-LE is only initialized once. We follow the convention of the decadal prediction community¹⁰ and refer to the former as the “initialized” ensemble and the latter as the “uninitialized” ensemble.”

More importantly, in the Methods section where the runs are described, considerably more care should be taken in explaining the different runs. While I read this section several times, I still don't think I understand. It is my understanding from the text that the CESM-DPLE runs are initialized as follows.

The atmospheric temperature portion is created somehow by round-off perturbations. What does that mean? $1e-6$ relative changes in values, randomly at all points, or with some assumed spatial or temporal decorrelation? Are these $1e-16$ changes? These changes seem negligible to me, but perhaps they are significant.

We clarified this in the methods at L204-206:

“An ensemble of 40 forecast members was created by making Gaussian perturbations to the initial atmospheric temperature field (order 10^{-14} K) at each grid cell. Ensemble spread in all other fields and model components developed as a result of the spread in the atmospheric state.”

Although the 10^{-16} K perturbations might seem negligible, they give rise to relatively rapid divergence in trajectories due to the system’s sensitive dependence on initial conditions. Here is an example of the divergence of individual CESM-LE members for daily sea surface temperature and vertical velocity at 50m in the CCS-LME. The divergence of trajectories results solely from the 10^{-14} K perturbation in atmospheric temperature (the same methodology used for CESM-DPLE initializations).

Fig R1. Evolution of area-weighted SST in the California Current for 40 individual members of CESM-LE (individual lines). Divergence of trajectories arises solely from the 10^{-14} K Gaussian perturbations made to the atmosphere component of the model in 1920 (day 0 in this plot).

Fig R2. Evolution of area-weighted vertical velocity (upwelling) at 50 m depth in the California Current for 40 individual members of CESM-LE (individual lines). Divergence of trajectories arises solely from the 10^{-14} K Gaussian perturbations made to the atmosphere component of the model in 1920 (day 0 in this plot).

Also confusing: on line 205, it mentions the round-off perturbations to temperature, but then shifts to the ocean-sea ice model and then shifts back to the atmospheric model on line 209.

We added this line following L205-206 to clarify that the spread in the ocean-sea ice model results from the perturbations to the atmospheric temperature:

“Ensemble spread in all other fields and model components developed as a result of the spread in the atmospheric state.”

The text says the atmosphere is initialized from a single ensemble member. Is that member randomly chosen? Would the ensemble mean be better?

The single ensemble member of CESM-LE that initializes CESM-DPLE was arbitrarily chosen to be member number 34. This is preferred over the ensemble mean, as the ensemble mean would suppress variability by smoothing out anomalies.

We updated L208-210 to the following:

“The atmosphere and land components were initialized from the November 1st restart files of a single arbitrary member of CESM-LE (ensemble member 34)¹¹.”

It would be helpful to describe the atmospheric component in full before introducing the ocean-sea ice component (line 207).

We reordered the structure, explaining atmosphere and land component initializations and the description of the model components prior to L207:

“The atmosphere and land components were initialized from the November 1st restart files of a single arbitrary member of CESM-LE (ensemble member 34)¹¹. The atmosphere component is the Community Atmosphere Model, version 5 (CAM5) with a finite-volume dynamical core at nominal 1° resolution and 30 vertical levels^{10,12}. Details on the land component can be found in Table S2.”

The ocean-sea ice values in re-initialization comes from the reconstruction. Seems ok, except that in Figure 1B, the blue dots (forecast means) don't sit on top of the black line (reconstruction). Why is that?

The re-initialized forecasts are initialized on November 1st at midnight for each year from the model reconstruction. Our “annual” predictability is assessed over the following January–December (clarified in L258–260). So the black line in this case is the January–December average of the following year and the blue dot is the January–December CESM-DPLE ensemble mean prediction for the following year. We would only expect the blue dot to sit exactly atop the black line if we plotted the November 1st values, due to the rapid divergence of trajectories shown in the earlier plots in this response.

We’ve added this clarification to the Figure 1 caption:

“The blue dots do not sit exactly atop the black line due to the rapid divergence of forecasts away from initialization within weeks.”

I assume that the BEC initialization also comes from the reconstruction, though this initialization is not stated anywhere.

Thank you for catching this. You are correct. We have added this clarification to the methods:

“The ocean (including biogeochemistry) and sea ice model components in CESM-DPLE were re-initialized from the November 1st restart files of a forced ocean-sea ice reconstruction (referred to as the “reconstruction”; see following section for configuration details).”

Please follow the spirit of this point by rewriting or significantly improving the Methods section so that the distinction between CESM-LE, CESM-DPLE, reconstruction, and persistence are crystal clear.

Thank you for your careful attention to our methods section. We feel that we’ve substantially rewritten it (per the above responses) to distinguish the model configurations more clearly.

2) Looking at Figure 1, it seems plausible that the blue lines are better predictors of the red lines than persistence of the reconstruction and than the un-re-initialized run (CESM-LE), but this is not shown. Can you add a line corresponding to the CESM-LE run to Figure 1 without over-cluttering it?

Thank you for this suggestion. We have added a line for the CESM-LE ensemble mean to Figure 1. We have labeled it “external forcing” for clarity in the figure and have explained in the caption that this is the CESM Large Ensemble mean.

Fig. 1. *CESM-DPLE experimental design.* (A) Trended and (B) detrended area-weighted annual surface pH anomalies for the (black) reconstruction, (red) observational product, (orange) CESM-LE ensemble mean, and (blue) CESM-DPLE decadal forecasts initialized in 1954, 1965, 1977, 1991, 2003, and 2017 (other initializations were omitted for visual clarity). The dark blue line is the ensemble mean forecast, and thin blue lines are the individual 40 forecasts. The blue dots do not sit exactly atop the black line due to the rapid divergence of forecasts away from initialization within weeks. The dashed red lines denote when the model loses observed variability in atmospheric CO₂ forcing (see Fig. S1A). The inset shows the California Current Large Marine Ecosystem bounds, over which all area-weighted analyses are computed.

3) Is there any estimate of the uncertainty in the observational estimates? The agreement between the area weighted reconstruction and the observational estimates suggests that the variability is real, but there's probably considerable uncertainty in the red-line (and individual points that go into Fig 4 calculations) that reduce significance. It's possible that you've already accounted for this, but I don't see discussion of it.

Unfortunately, the JMA Ocean CO₂ map product does not offer uncertainty for the pH values directly. They do offer estimates of uncertainty in pCO₂¹³ and alkalinity¹⁴, which were used to solve for their pH product. The uncertainty was reported as the root mean square error (RMSE) of the gridded estimate compared to the *in situ* observations. They report an RMSE of 10-20 μatm for pCO₂ in the mid-latitude northern hemisphere and 8.1 $\mu\text{mol kg}^{-1}$ for surface alkalinity relative to the PACIFICA campaign.

We added the following to the observational description in the methods:

“There are no uncertainty estimates available for the pH product, but the authors report a root mean square error (gridded estimate compared to in situ observations) of 10-20 μatm for pCO₂ in the northern hemisphere mid-latitudes and 8.1 $\mu\text{mol kg}^{-1}$ for surface alkalinity relative to the PACIFICA campaign^{14,15}. Note that the global average JMA surface pH is within the uncertainty of the SOCAT-based estimate for all years (Fig. S9).”

Fig S9. Validation of JMA observational product against the SOCAT-based product. Global mean surface pH from the SOCAT-based product (black) and JMA observational product (red) used in this study. Gray shading denotes uncertainty bounds provided with the SOCAT-based product.

4) There is probably not space for this, but since you make a prediction for 2020 being more basic at the coast and more acidic offshore, can you explain that signal more? At first I was surprised, simply assuming that the coastal anomaly should always be more acidic than the area-weighted mean. But your later results suggest that my first interpretation was wrong, that it's the subsurface DIC anomaly that is anomalously low prior to upwelling, causing a reduced near coast signal in 2020? An alternative might be reduced alongshore winds in 2020 or a change in the timing of the onset of coastal upwelling? Given your prediction, it would be nice if you can trace it back to the driver within the model. The obvious place to do this would be on the section discussing the significance of sDIC.

We appreciate your interesting comments on this figure. We agree that if we were to include the 2020 forecast, we should spend considerable time investigating what in particular caused such a forecast in the CESM-DPLE. We feel that the analysis of a specific forecast like this could constitute its own letters-style paper. Due to space limitations, we have removed panel C from Figure 1.

Reviewer #3 (Remarks to the Author):

The study by Brady and Coauthors discusses a new application of an ocean forecast system to predict the progression of ocean acidification (specifically, surface pH) in the California Current System (CCS). They show that the forecast is more skillful than naive “persistence” forecasts, and is able to capture some of the variability in acidification several year in advance.

Ocean acidification is one of the main anthropogenic stressors threatening marine ecosystems, and has received increasing attention over the past decade or so. Ocean biogeochemistry forecasting is still at the early stages, with several interesting results recently published (e.g. Park et al., 2019, Science). This paper merges the two topics into a coherent framework, and perhaps represent the first example, to my knowledge, of acidification forecasting on multi-year timescales. (Skillful forecasts of other carbon cycle processes, e.g. carbon uptake, have been presented in other studies.)

The paper is well written, and is clearly the result of a significant amount of work, in particular an impressive amount of simulations with a state-of-art Earth System Model (ESM) ensemble “machinery”. As such, I think the paper is definitely publishable. However, I have some concerns that make me question publication in Nature Communication. These are mostly related to the ability of the ESM to provide credible simulation (and forecast) of the patterns of acidification at the local scales that are relevant for management of marine resources. Thus, I think that the potential of the forecast described in the paper to provide predictions relevant to management is inflated. My impression is that the paper could be more suitable to a field-specific journal, as a first step towards useful regional predictions of acidification or biogeochemistry.

We thank the anonymous referee for their careful review of our manuscript which helped to substantially improve it.

Main Comments:

- The approach is based on an ESM with a horizontal resolution of 100km. This is insufficient to capture the spatial gradients of acidification (pH, saturation state) that naturally occurs along the CCS, both cross-shore and alongshore. These are the relevant gradients for many important fisheries in the CCS, in particular the ones that are more susceptible to acidification (e.g. Dungeness crabs, shellfish), which are very coastal and confined to the shelf. I have a hard time considering a 100-km resolution forecast relevant in an oceanographic region dominated by circulation and gradients at scales of order of few 10s of km, i.e. the coastal upwelling band, the shelf, and the alongshore heterogeneity driven by topography. There has been a fairly long list of papers showing how capturing these scales in the ocean (and even atmosphere, as forcing) is important to reproduce realistic physical and biogeochemical gradients in upwelling systems. Some of these studies have shown that ocean model resolution really matters in the CCS, with coarse models unable to represent realistic spatial patterns in physical and C cycle variables. For example, Fiechter et al., 2014, Global Biogeochemical Cycles, showed that even models at 30km resolution are dramatically biased in their representation of surface pCO₂ (closely related to pH), compared to models at resolution less than 10km. Other work has shown that factors like the

shape of winds at scales of ~10km control the strength of coastal vs. offshore upwelling, determining the redistribution of nutrient- and DIC-rich waters (Renault et al., 2018, Nature Geosciences). Interactions of winds and currents with coastal topographic features at scales of 10s-100km also shape productivity, surface chlorophyll, and presumably C-system parameters and pH (Fiechter, et al., 2018, Global Biogeochemical Cycles). Similarly, the coastal undercurrent that carries DIC-rich waters from the tropics just below the surface along the coast, and serves as a water reservoir for coastal upwelling, is not represented at scales of 100km. In summary, the tools utilized in the paper seem more appropriate for a large-scale analysis, rather than a coastal analysis that could be relevant for marine resource management. The Authors are of course aware of these shortcomings, and discuss them in the manuscript; however, they are important, and they limit the ability of the forecast to provide information at the relevant scales.

We agree with your assessment that the 100 km grid resolution does not accurately depict nearshore processes, like the strong coastal upwelling-driven outgassing near the coastline. This is what drove our decision to do a large-scale analysis of the California Current LME, rather than just the coastal region.

We have added a sentence to point this out following L91-92:

“We focus on the entire Large Marine Ecosystem, since the 1° x 1° model grid cannot resolve the coastal upwelling of corrosive waters that occurs on scales smaller than the grid resolution.”

We have also updated the abstract to reflect that our results are applicable on large scales in the CCS:

“Our results demonstrate the potential for ESMs to provide predictions relevant to managing the onset and impacts of ocean acidification on large scales in the CCS.”

And to the discussion:

“Although these forecasts cannot aid directly in the management of coastal fisheries at this spatial resolution, our results demonstrate the feasibility of making skillful surface pH predictions on multiannual to decadal timescales.”

We moved the “Model Evaluation” section from the methods to the main text and expanded it significantly to address the grid resolution shortcomings and to validate the coarser model resolution against the observational product:

“Previous evaluations of the physical circulation and carbonate chemistry in the version of CESM used for CESM-DPLE suggest that, despite the relatively coarse 1° x 1° model grid, CESM provides a good fit to observational climatologies of alongshore wind stress, surface pCO₂, and air–sea CO₂ fluxes in the CCS^{1,2}. Modeled alongshore wind stress—the primary driver of coastal upwelling—closely matches the magnitude and seasonality of observations, with peak upwelling-favorable conditions spanning April to September¹.

The large-scale spatial structure of air–sea CO₂ fluxes in the model exhibits poleward CO₂ uptake and equatorward CO₂ outgassing, matching that of modern observationally based estimates^{2,3}. Importantly, we note that CESM cannot capture the nearshore outgassing of CO₂ associated with the coastal upwelling of carbon-enriched waters that occurs on a scale smaller than the resolution of the model grid^{2,4}. The modeled monthly climatology of area-weighted surface ocean pCO₂ in the CCS closely resembles that of the observationally based estimate, due to the model’s proper simulation of the magnitude and phasing of thermal (solubility-driven) and non-thermal (circulation- and biology-driven) pCO₂ effects^{2,3}.”

- Perhaps as a consequence of the coarse resolution, the acidification signals that the paper show to be predictable on multi-year time scales are surprisingly small. As such, it is not clear that they would have a significant influence on the ecosystem, or that they would be useful for management decisions. For example, Fig. 1b-c indicates ability of predicting fluctuations on the order of 0.003 pH units. This are very small variations (roughly corresponding to a change in H⁺ concentration of less than 1%!). My sense is that the type of pH variation that could drive significant ecological changes (e.g. changing the saturation state of carbonates by a significant amount, e.g. on the order of 0.1 saturation units) would need to be at least an order of magnitude larger, if not more. The paper does not discuss in detail the potential importance for ecosystem, and hence for management, of the type of fluctuations that are shown to be predictable. This should be addressed to make a strong case for the usefulness of the multi-year pH forecast for applications. The Authors should strive to convince the readers that the predicted perturbations actually matter for ecosystem and management, as claimed in the introduction.

We agree that the variability being predicted in pH is small due to the coarse resolution and area-weighting over the full CCS LME. However, because these pH fluctuations are co-dependent with other carbonate chemistry parameters on variability in environmental drivers such as SST, we find that they are associated with fairly large fluctuations in variables such as omega aragonite. Here, pH varies by only 0.002 units, but CO₃, sDIC, and omega aragonite vary by 2.6 mmol/m³, 4.4 mmol/m³, and 0.04 units, respectively. They also experience swings on the order of 5 mmol/m³, 10 mmol/m³, and 0.1 units, respectively. Note that we chose to focus on predictability in surface pH in particular, rather than other carbonate chemistry variables, since we have a gridded observational product with which to evaluate predictive skill.

We added the following to the discussion to reflect this (as well as supplemental Fig. S7):

“However, the coarser grid resolution suppresses variability in surface pH. In turn, the annual surface pH anomalies being predicted are smaller than 0.01 units (Fig. 1B), but these relatively small anomalies are associated with large fluctuations in other environmentally relevant variables, such as the aragonite saturation state, which varies on the order of 0.1 units (Fig. S6D).”

Fig S6. *Interannual variability of carbonate chemistry in the California Current. (A)* Area-weighted annual averages of surface pH in the model reconstruction in the California Current Large Marine Ecosystem after removing a second-order polynomial fit. **(B-D)** As in (A), but for the carbonate ion, salinity-normalized DIC, and the aragonite saturation state, respectively.

We also changed our accuracy metric from MAE to NMAE, the normalized Mean Absolute Error. This normalizes the MAE by the interannual variability from the verification product (e.g. the model reconstruction or JMA observations). This shows that our predictions of both the reconstruction and JMA observations fall within the interannual variability, despite it being relatively small. We added the following to the results and discussion, as well as updated Fig. 3 and added a new figure to the main text:

“The NMAE is smaller than both persistence and the uninitialized forecast over all ten lead years, and falls within the magnitude of surface pH interannual variability in the model reconstruction (Fig. 4B).”

“In spite of the relatively small target anomalies being predicted, CESM-DPLE forecast error (as measured by the NMAE) falls within the spread of the historical surface pH variability (Fig. 4B and 6).”

Fig. 4. Area-weighted potential predictability of surface pH in the California Current and driver variables of surface pH predictability. (A) ACCs for ten lead years for (blue) CSM-DPLE, (black) a persistence forecast from the reconstruction, and (grey) the uninitialized CSM-LE ensemble mean. Filled circles denote statistically significant positive correlations at the 95% level using a t test. An effective sample size is used in the t test to account for autocorrelation in the two time series being correlated⁵. The critical value required for a statistically significant correlation ranges from 0.26 to 0.32 across leads, as computed by inverting the t statistic formula. Black and gray asterisks indicate significant predictability over persistence and the uninitialized forecast at the 95% level using a z test, respectively. (B) As in (A), but for NMAE and without significance testing. Values below (above) 1 indicate that the forecast falls within (outside of) the interannual variability of surface pH in the reconstruction. (C) Scaled predictability in common pH units (see methods) of (black) sea surface salinity, (teal) sea surface temperature, (gold) salinity-normalized alkalinity, and (red) salinity-normalized dissolved inorganic carbon.

Fig. 6. Normalized mean absolute error of initialized surface pH anomaly forecasts relative to observations in the California Current. (A to E) NMAE of CESM-DPLE initialized forecast of detrended annual surface pH anomalies for lead years one through five relative to the observational product over 1990–2005. (F to J) NMAE of a persistence forecast for the observations for lead years one through five. Purple colors (values below one) indicate that the forecast error falls within the interannual variability of observations; orange colors (values above one) fall outside of the interannual variability of observations. (K to O) Difference between the CESM-DPLE forecast and observational persistence NMAEs. Green colors indicate that the initialized forecast have lower error than the persistence forecast.

- The main signal in interannual variations of pH and C cycle tracer is a forced signal driven by the atmospheric pCO₂ increase (as discussed for example in Gruber et al., 2012, Science). This is a predictable, major signal, and yet is by construction discarded by the Authors. I understand that predicting fluctuations driven by internal dynamics is a much more challenging (and dynamically interesting) problem that requires a major ESM-based forecast machinery. But nonetheless, removing this large, multi-annual signal forces the Authors to focus on small residual perturbations (Fig. 1b-c) that are probably not as impactful.

We agree that it was an oversight to not include predictability results *with* the forced ocean acidification signal. This illustrates that our predictions still beat out persistence with the dynamical forecasting system. We have added the following paragraph to the discussion and added a supplemental figure:

“We focus on assessing predictability in surface pH after removing the ocean acidification trend to highlight the role of initialization in engendering predictability. Our results are similar if we conduct the analysis on trended surface pH (Fig. S8).”

Fig S8. Area-weighted predictability of trended surface pH forecasts in the California Current. (A) ACCs for surface pH forecasts relative to the model reconstruction over ten lead years for (blue) CESM-DPLE, (black) a persistence forecast from the reconstruction, and (grey) the uninitialized CESM-LE ensemble mean. Filled circles denote statistically significant positive correlations at the 95% level using a t test. An effective sample size is used in the t test to account for autocorrelation in the two time series being correlated¹. Black and gray asterisks indicate significant predictability over persistence and the uninitialized forecast at the 95% level using a z test, respectively. (B) As in (A), but for surface pH forecasts relative to the observational product.

Additionally, member-to-member anomalies (thin lines in Fig. 1b) are substantially larger than the model ensemble mean anomaly (as are reconstructions and observations), suggesting that it is really poorly- or non-predictable fluctuations that could have a greater impact on pH, and possibly on ecosystem. Larger, shorter fluctuations could be driven by variability that has a weak “memory”, e.g. winds fluctuations driving anomalous advection or upwelling. The existence of larger, unpredictable fluctuations also reduces the direct utility of the multi-annual forecast shown in the paper. That said, even these individual-member fluctuations appear quite minor when compared to the multi-year progression of acidification in Fig. 1a, which is driven by the atmospheric pCO₂ increase.

We agree that there is further interesting information encoded in the individual ensemble members that we didn't make use of by using deterministic metrics. We chose deterministic metrics for this study, since surface pH is a continuous variable without clear "event" thresholds. We think that future studies could leverage probabilistic metrics (e.g. Brier score) to assess predictability in variables such as omega, with a clear "event" threshold of 1.

- On a related note, I wonder to what extent potentially-damaging acidification events occur at the spatiotemporal scale of upwelling events, which are far less predictable than the broader-scale, more persistent conditions that are addressed in this study.

We agree that this would be an interesting study. However, it depends on a model experiment in line with subseasonal-to-seasonal (S2S) protocol, which typically re-initializes model forecasts every single week and generates daily forecasts with lead times of 1-2 months. The finest temporal resolution we have available for this study is monthly.

- Finally, the paper feels a bit short on broader implications, and the case for marine resource managers or other stakeholders to use this type of multi-annual forecasts is weak (e.g. in the introduction and discussion). As a comparison, the paper by Park et al. 2019, Science, was able to connect predictability in SST and Chl to fishery catch predictions (albeit on shorter timescales), which has more obvious direct implications.

As mentioned in earlier responses, we have rewritten bold language in the introduction and discussion about regional fisheries management. We have instead focused on the fact that our study demonstrates the feasibility of making skillful pH predictions on these time scales and that the CESM-DPLE has utility as boundary conditions to improve regional forecasts.

Added to the abstract:

"Initialized ESMs could also provide boundary conditions to improve high-resolution regional forecasting systems."

Added to the discussion:

"Our results demonstrate for the first time the potential for an initialized ESM to retrospectively predict surface pH multiple years in advance in a complex, sensitive, and economically important oceanic region. Although these forecasts cannot aid directly in the management of coastal fisheries at this spatial resolution, our results demonstrate the feasibility of making skillful surface pH predictions on multiannual to decadal timescales. Further, global initialized ESM forecasts can be used as boundary conditions to improve existing regional biogeochemical forecasting and extend their lead times."

- That said, my sense is that the work presented by Brady et al. is significant, and could be indeed useful in connection to resource management, for example to help downscaled biogeochemical predictions. For example, it could be used to initialize and drive regional

biogeochemical forecast systems, of which a few are being developed along the US West Coast (e.g. Siedlecki et al., 2016, Sci. Rep., etc.). But this is beyond what is discussed in the manuscript.

Thank you very much for this suggestion. We feel that this is a great addition the manuscript and have added comments on using the CESM-DPLE as boundary conditions accordingly (see previous response).

Other comments:

- Line 28: I am not sure pteropods, coccolithophores, and shellfish are “keystone” species in the CCS. This statement should be better supported. Likewise, the most valuable commercial fisheries in California today are Dungeness crab and market squid, and I am not sure how either connect to the species listed.

We have reframed the sentence in L28 to focus on the fact that shellfish contribute directly to this revenue:

“These conditions adversely affect a wide range of organisms that precipitate calcium carbonate shells, such as pteropods, coccolithophores, and shellfish¹⁶. Shellfish in particular contribute significantly to the \$6B in revenue per year provided by commercial and recreational fisheries in the CCS¹⁷.”

- Lines 33-34. Shellfisheries tend to be fairly coastal, and affected by small-scale (e.g. coastal upwelling) processes. Also, I am not sure buffering fishery sites with sodium carbonate is something that managers realistically consider, given costs, scale, effectiveness, and legal implications.

We have removed these lines from the introduction, since they address more coastal phenomena that we do not resolve at this grid resolution.

- Lines 42-50. Maybe more details on what management approaches have indeed included information on multi-annual to decadal timescales would help the argument. To my knowledge, most fishery assessments are based on annual projections. Also, I am not aware of fishery management that includes ocean biogeochemistry variables. Examples would help.

We have rewritten this section to remove the suggestion that fisheries managers use persistence forecasts. Instead we focused on defining persistence forecasts as a baseline for assessing skill in initialized predictions:

“However, no studies have attempted to predict ocean biogeochemistry in the CCS at the multiannual to decadal scale, as decadal forecasting of ocean biogeochemistry is still in its infancy¹⁸⁻²². This temporal scale is critical for fisheries managers, as it aids them in setting annual catch limits, changing and introducing closed areas, and adjusting quotas for internationally shared fish stocks⁶. Some level of skill can be provided by persisting anomalies from year to year in the system⁶. These so-called persistence forecasts are

commonly used as a reference to put initialized skill into context and work at lead times commensurate with the decorrelation timescales of the system⁶⁻⁹.”

- Lines 130-136. This is a cool result, but again the forecast signal has a pretty small magnitude.

We hope that our earlier discussion of this alleviates this issue.

- Fig. 4. Why is the coastal band omitted in these figures? Is it missing from the observational database? Unfortunately, important marine resources are distributed along this band.

Correct. The observational product masked out marginal seas, coastal zones, and the Arctic ocean¹³ (see the below figure). We agree this is unfortunate, but we are grateful to have a gridded product at all to evaluate predictive skill across the rest of the LME.

Fig R3. Mean surface pH from the JMA observational product over 1990-2018. The white band in the nearshore region is masked out by the data providers.

- Line 150 and Fig. S4. This is also a cool analysis.

Thank you!

- Lines 231-246. I am confused by this section. Is this a description of the JMA product, or was the calculation re-done as part of this paper? This should be clarified.

This is a consolidated description of the JMA product. The authors did not perform any additional post-processing (outside of statistical analyses for this manuscript) on the product. We clarified this following L234:

“Here, we describe the key steps followed by the authors of the JMA product to derive their surface pH estimates.”

- Line 280. “Increase to infinity”. This seems extreme. To go to infinity, either the forecast or the verification need to go to infinity, which seems unphysical.

While this is an accurate description of the math, we agree it is unnecessary in this situation. We have replaced this metric with normalized mean absolute error (NMAE) and have removed the reference to infinity in its description:

“To quantify the magnitude of forecast error, or the accuracy in our forecasts, we use the normalized mean absolute error²³ (NMAE), which is the MAE normalized by the interannual standard deviation of the verification data. The NMAE is 0 for perfect forecasts, less than 1 when the forecast error falls within the variability of the verification data, and increases as the forecast error surpasses the variability of the verification data. MAE is used instead of bias metrics such as the root mean square error (RMSE), as it is a more accurate assessment of bias in climate simulations²⁴.

$$NMAE(\tau) = \frac{1}{N} \sum_{\alpha=1}^N \frac{|F'_{\alpha}(\tau) - O'_{\alpha+\tau}|}{\sigma_{O'}(\tau)}$$

Where N is the number of initializations and $\sigma_{O'}$ is the standard deviation of the verification data over the verification window.”

References

1. Brady, R. X., Alexander, M. A., Lovenduski, N. S. & Rykaczewski, R. R. Emergent anthropogenic trends in California Current upwelling. *Geophys. Res. Lett.* **44**, 2017GL072945 (2017).
2. Brady, R. X., Lovenduski, N. S., Alexander, M. A., Jacox, M. & Gruber, N. On the role of climate modes in modulating the air–sea CO₂ fluxes in eastern boundary upwelling systems. *Biogeosciences* **16**, 329–346 (2019).
3. Landschützer, P. *et al.* A neural network-based estimate of the seasonal to inter-annual variability of the Atlantic Ocean carbon sink. *Biogeosciences* **10**, 7793–7815 (2013).
4. Laruelle, G. G. *et al.* Global high-resolution monthly pCO₂ climatology for the coastal ocean derived from neural network interpolation. *Biogeosciences* **14**, 4545–4561 (2017).
5. Bretherton, C. S., Widmann, M., Dymnikov, V. P., Wallace, J. M. & Bladé, I. The Effective Number of Spatial Degrees of Freedom of a Time-Varying Field. *J. Clim.* **12**, 1990–2009 (1999).

6. Tommasi, D. *et al.* Managing living marine resources in a dynamic environment: The role of seasonal to decadal climate forecasts. *Prog. Oceanogr.* **152**, 15–49 (2017).
7. Stock, C. A. *et al.* Seasonal sea surface temperature anomaly prediction for coastal ecosystems. *Prog. Oceanogr.* **137**, 219–236 (2015).
8. Jacox, M. G., Alexander, M. A., Stock, C. A. & Hervieux, G. On the skill of seasonal sea surface temperature forecasts in the California Current System and its connection to ENSO variability. *Clim. Dyn.* (2017) doi:10.1007/s00382-017-3608-y.
9. Hervieux, G. *et al.* More reliable coastal SST forecasts from the North American multimodel ensemble. *Clim. Dyn.* (2017) doi:10.1007/s00382-017-3652-7.
10. Yeager, S. G. *et al.* Predicting Near-Term Changes in the Earth System: A Large Ensemble of Initialized Decadal Prediction Simulations Using the Community Earth System Model. *Bull. Am. Meteorol. Soc.* **99**, 1867–1886 (2018).
11. Lovenduski, N. S., Bonan, G. B., Yeager, S. G., Lindsay, K. & Lombardozzi, D. L. High predictability of terrestrial carbon fluxes from an initialized decadal prediction system. *Environ. Res. Lett.* **14**, 124074 (2019).
12. Hurrell, J. W. *et al.* The Community Earth System Model: A Framework for Collaborative Research. *Bull. Am. Meteorol. Soc.* **94**, 1339–1360 (2013).
13. Iida, Y. *et al.* Trends in pCO₂ and sea–air CO₂ flux over the global open oceans for the last two decades. *J. Oceanogr.* **71**, 637–661 (2015).
14. Takatani, Y. *et al.* Relationships between total alkalinity in surface water and sea surface dynamic height in the Pacific Ocean. *J. Geophys. Res. Oceans* **119**, 2806–2814 (2014).
15. Iida, Y. *et al.* Trends in pCO₂ and sea–air CO₂ flux over the global open oceans for the last two decades. *J. Oceanogr.* **71**, 637–661 (2015).

16. Doney, S. C., Fabry, V. J., Feely, R. A. & Kleypas, J. A. Ocean Acidification: The Other CO₂ Problem. *Annu. Rev. Mar. Sci.* **1**, 169–192 (2009).
17. Fisheries economics of the United States 2015. (2017).
18. Seferian, R. *et al.* Multiyear predictability of tropical marine productivity. *Proc. Natl. Acad. Sci.* **111**, 11646–11651 (2014).
19. Li, H., Ilyina, T., Müller, W. A. & Sienz, F. Decadal predictions of the North Atlantic CO₂ uptake. *Nat. Commun.* **7**, (2016).
20. Séférian, R., Berthet, S. & Chevallier, M. Assessing the Decadal Predictability of Land and Ocean Carbon Uptake. *Geophys. Res. Lett.* **45**, 2455–2466 (2018).
21. Lovenduski, N. S., Yeager, S. G., Lindsay, K. & Long, M. C. Predicting near-term variability in ocean carbon uptake. *Earth Syst. Dyn.* **10**, 45–57 (2019).
22. Li, H., Ilyina, T., Müller, W. A. & Landschützer, P. Predicting the variable ocean carbon sink. *Sci. Adv.* **5**, eaav6471 (2019).
23. Jolliffe, I. T. & Stephenson, D. B. *Forecast verification: a practitioner's guide in atmospheric science.* (John Wiley & Sons, 2012).
24. Willmott, C. & Matsuura, K. Advantages of the mean absolute error (MAE) over the root mean square error (RMSE) in assessing average model performance. *Clim. Res.* **30**, 79–82 (2005).

REVIEWERS' COMMENTS:

Reviewer #1 (Remarks to the Author):

This is a significantly improved version of the original manuscript and the authors have considered my previous comments. I think the model-data evaluation brings valuable information in the manuscript and support the key findings of the authors team. I believe this paper will be an important contribution to the scientific literature.

Reviewer #2 (Remarks to the Author):

The authors have done a good job of addressing my comments. The paper is interesting, sound in its analysis, and reads well. I recommend publication.

Reviewer #3 (Remarks to the Author):

This is my second review of the manuscript by Brady and Coauthors. As outlined in my previous assessment, the paper addresses a topic of fairly broad interest (ocean acidification), and is the result of a substantial amount of work with state-of-art tools in ocean multi-year forecasting. With this work, the Authors do a credible job in demonstrating potential multi-annual predictability of large-scale pH anomalies over the California Current region. In themselves, the results appear to be robust and supported by a well-designed study and analysis.

The revised manuscript is significantly improved from the previous version, with a more substantial assessment and validation of the model and the forecasts, and a more nuanced discussion. My suggestions and those of two other Reviewers have also been addressed, generally in a satisfactory way.

That said, two criticisms of the study remain, namely, that the scales at which the prediction is performed (and credible) are likely too broad to capture signals and dynamics that are relevant for management of coastal marine resources more directly impacted by acidification, and that the interannual signals that are predictable likely have a limited biological impact, especially when they are couched within the larger (and more predictable) variability of the seasonal cycle and of the forced anthropogenic acidification.

The Authors have corrected some of the sentences that overstated the implications for management, for example they note in the discussion that their forecast "cannot aid directly in the management of coastal fisheries at this spatial resolution". This is a more balanced assessment, but it highlights a limitation for the implications of the study. While it demonstrates predictability of pH anomalies on multi-annual timescales, how this predictability can be directly useful is not as obvious. Thus, the study would likely be of interest to a community focusing on the ocean C cycle and its interannual predictability, but it may not be equally relevant to a wider readership.

Response to Reviewers

Reviewer #1 (Remarks to the Author):

This is a significantly improved version of the original manuscript and the authors have considered my previous comments. I think the model-data evaluation brings valuable information in the manuscript and support the key findings of the authors team. I believe this paper will be an important contribution to the scientific literature.

Thank you again for your time and helpful comments. We agree that the new model-data evaluation section adds a lot to the paper, and we enjoyed doing the additional analysis.

Reviewer #2 (Remarks to the Author):

The authors have done a good job of addressing my comments. The paper is interesting, sound in its analysis, and reads well. I recommend publication.

Thank you again for your impactful comments and time in reviewing our paper. We appreciate the kind words and feel that the manuscript was substantially improved as a direct result of your review.

Reviewer #3 (Remarks to the Author):

This is my second review of the manuscript by Brady and Coauthors. As outlined in my previous assessment, the paper addresses a topic of fairly broad interest (ocean acidification), and is the result of a substantial amount of work with state-of-art tools in ocean multi-year forecasting. With this work, the Authors do a credible job in demonstrating potential multi-annual predictability of large-scale pH anomalies over the California Current region. In themselves, the results appear to be robust and supported by a well-designed study and analysis.

The revised manuscript is significantly improved from the previous version, with a more substantial assessment and validation of the model and the forecasts, and a more nuanced discussion. My suggestions and those of two other Reviewers have also been addressed, generally in a satisfactory way.

Thank you for your time in re-reviewing our manuscript. We appreciate the comments and are happy that our edits from the previous submission are satisfactory.

That said, two criticisms of the study remain, namely, that the scales at which the prediction is performed (and credible) are likely too broad to capture signals and dynamics that are relevant for management of coastal marine resources more directly impacted by acidification, and that the interannual signals that are predictable likely

have a limited biological impact, especially when they are couched within the larger (and more predictable) variability of the seasonal cycle and of the forced anthropogenic acidification.

The Authors have corrected some of the sentences that overstated the implications for management, for example they note in the discussion that their forecast “cannot aid directly in the management of coastal fisheries at this spatial resolution”. This is a more balanced assessment, but it highlights a limitation for the implications of the study. While it demonstrates predictability of pH anomalies on multi-annual timescales, how this predictability can be directly useful is not as obvious. Thus, the study would likely be of interest to a community focusing on the ocean C cycle and its interannual predictability, but it may not be equally relevant to a wider readership.

In coordination with the editor, we have further toned down management language to address these concerns. In the abstract, we change the original sentence,

“Our results demonstrate the potential for ESMs to provide predictions relevant to managing the onset and impacts of ocean acidification at large scales in the CCS.”

to the following:

“Our results demonstrate the potential for ESMs to provide skillful predictions of ocean acidification on large scales in the CCS.”

We also added another sentence to the Discussion to further drive home the point that our coarser ESM forecasts cannot be used directly in management, but that our paper shows the potential for them to improve regional forecasting systems which are run at a resolution much more fitting for management decisions. The bolded text below reflects our additions.

*“Although these forecasts cannot aid directly in the management of coastal fisheries at this spatial resolution, our results demonstrate the feasibility of making skillful surface pH predictions on multiannual to decadal timescales. Further, **our work suggests that** global initialized ESM forecasts can be used as boundary conditions to improve existing regional biogeochemical forecasting and to extend their lead times. **Dynamically downscaled decadal forecasts with high resolution regional models could go a long way toward improving fisheries management in sensitive coastal regions on interannual timescales.**”*

Thank you again for your careful review of this manuscript. We feel that it was improved significantly by incorporating your comments and concerns.